# Patient perspectives on interpersonal aspects of healthcare and patient-centeredness at primary health facilities: A mixed methods study in rural Eastern Uganda

Everlyn Waweru [1,2,3]*, Tom Smekens [1], Joanna Orne-Gliemann[2], Freddie Ssengooba[4], Jacqueline Broerse[3], Bart Criel[1]

1 Department of Public Health–Health Systems and Equity Unit, Institute of Tropical Medicine, Antwerp, Belgium, 2 Population Health Department, University of Bordeaux, Bordeaux, France, 3 Department of Public Health–Quality of Care, Athena Institute, Faculty of Science, Vrije University, Amsterdam, Netherlands, 4 Department of Health Policy Planning & Management, Makerere University College of Health Sciences, Kampala, Uganda

* ewaweru@itg.be

**Data Availability Statement:** The anonymized and de-identified quantitative patient exit data set is provided as a supplementary file and quotes

## Abstract

### Introduction

Patient-centered care (PCC) is an approach to involve patients in health care delivery, to contribute to quality of care, and to strengthen health systems responsiveness. This article aims to highlight patient perspectives by showcasing their perceptions of their experience of PCC at primary health facilities in two districts in Uganda.

### Methods

A mixed methods cross-sectional study was conducted in three public and two private primary health care facilities in rural eastern Uganda. In total, 300 patient exit survey questionnaires, 31 semi-structured Interviews (SSIs), 5 Focus Group Discussions (FGDs) and 5 feedback meetings were conducted. Data analysis was guided by a conceptual framework focusing on (1) understanding patients' health needs, preferences and expectations, (2) describing patients perceptions of their care experience according to five distinct PCC dimensions, and (3) reporting patient reported outcomes and their recommendations on how to improve quality of care.

### Results

Patient expectations were shaped by their access to the facility, costs incurred and perceived quality of care. Patients using public facilities reported doing so because of their proximity (78.3% in public PHCs versus 23.3% in private PHCs) and because of the free services availed. On the other hand, patients attending private facilities did so because of their perception of better quality of care (84.2% in private PHCs versus 21.7% in public PHCs). Patients expectations of quality care were expressed as the availability of medication, shorter waiting times, flexible facility opening hours and courteous health workers.

relevant to this manuscript are provided in the text. Interview audio files and complete transcripts (which contain additional information and will be used for other manuscripts) are governed under the Institute of Tropical Medicine (ITM) data sharing and open access policy. Access requests for ITM research data can be made to ITM's central point for research data access by means of submitting a completed Data Access Request Form. These requests will be reviewed for approval by ITMs Data Access Committee, with further approval from the ITM Research Ethics Committee. Please see this link for more information https://www.itg.be/F/data-sharing-open-access. Contact information for ITM data access committee is eb.gti@sseccaatadhcraeserMTI.

**Funding:** This work was supported by funding from the European Commission, through the Erasmus Mundus Joint Doctorate Fellowship, Specific Grant Agreement 2016-1346, awarded to EW. The funders had no role in study design, data collection and analysis, decision to publish, or preparation of the manuscript.

**Competing interests:** The authors have declared that no competing interests exist.

**Abbreviations:** ART, Antiretroviral Therapy; CHW, Community Health Worker; DHMT, District Health Management Team; FGDs, Focus group discussions; HC, Health centre; HIV/AIDS, Human Immunodeficiency Virus infection / Acquired Immune Deficiency Syndrome; IMHDSS, Iganga Mayuge Health and Demographic Surveillance Site; LMICs, Low and middle income countries; MHC, Maternal and child health care; NCD, Non-Communicable Disease; NGO, Non-governmental organization; Nvivo, Qualitative data analysis computer software package produced by QSR International; PCC, Patient-centered care; PHC, Primary health care; PI, Principal investigator; RC, Routine care; SC, Specialized care; SSIs, Semi-structured Interviews; STATA, Software for statistics and data science; VHTs, Village health teams.

Analysis of the 300 responses from patients interviewed on their perception of the care they received, pointed to higher normalized scores for two out of the five PCC dimensions considered: namely, exploration of the patient's health and illness experience, and the quality of the relationship between patient and health worker (range 62.1–78.4 out of 100). The qualitative analysis indicated that patients felt that communication with health workers was enhanced where there was trust and in case of positive past experiences. Patients however felt uncomfortable discussing psychological or family matters with health workers and found it difficult to make decisions when they did not fully understand the care provided. In terms of outcomes, our findings suggest that patient enablement was more sensitive than patient satisfaction in measuring the effect of interpersonal patient experience on patient reported outcomes.

## Discussion and conclusion

Our findings show that Ugandan patients have some understanding of PCC related concepts and express a demand for it. The results offer a starting point for small scale PCC interventions. However, we need to be cognizant of the challenges PCC implementation faces in resource constrained settings. Patients' expectations in terms of quality health care are still largely driven by biomedical and technical aspects. In addition, patients are largely unaware of their right to participate in the evaluation of health care. To mitigate these challenges, targeted health education focusing on patients' responsibilities and patient's rights are essential. Last but not least, all stakeholders must be involved in developing and validating methods to measure PCC.

## Introduction

Involving patients in the planning, delivery and evaluation of healthcare has been endorsed as an important approach towards improving the quality of health care services and the responsiveness of health systems worldwide [1, 2]. This is even more crucial in resource constrained settings where historically, quality improvement strategies have been more focused on health care providers, with little or no attention to consumer perspectives in the design and assessment of quality improvement interventions [3–5]. Health care consumers in these settings have also been reported to have low expectations of what is good quality health care [6]. Consequently, patient-centered care (PCC) has been advocated as one of the ways in which patients can participate in health care.

PCC has been defined by the Institute of Medicine as care that is "respectful of, and responsive to, individual patient preferences, needs and values, and ensuring that patient values guide all clinical decisions" [7]. Barry et al. [8] similarly define PCC as being "about considering people's desires, values, family situations, social circumstances and lifestyles; seeing the person as an individual, and working together to develop appropriate solutions". Literature also describes PCC as "care where the patient is the source of control; care where knowledge is shared and information flows freely; care where transparency is necessary and the needs of the patient are anticipated" [1] and "care where patients are encouraged to participate in, and make decisions about their health and health care" [8, 9]. Established patient organizations view PCC as "patients acting as equal and informed partners in decision making", in contrast to traditional medical paternalism. It is partly about valuing patients as consumers of services,

who should be empowered through better information, greater choice, and the opportunity to give feedback and rate health care services. It is also about patients exercising their rights and responsibilities as citizens" [10, 11]. McCormack, Borg et al. recommend the patients' perspective on what constitutes high-quality care as central to the implementation and evaluation of PCC [12]. Several reviews conducted in developed countries with—more experience in the implementation of PCC—have also associated PCC approaches with increased satisfaction with care, improved patient enablement and functioning, and increased ability of caregivers to care for patients at home [13–18].

Positive outcomes have directed many health care organizations to consider consumer wants and needs as part of continuous quality improvement. Bechel *et al.* distinguished among the concepts of patient-focused care, patient-based care and patient-centered care [19]. They argue that *patient-focused care* emphasizes tailoring services to patient needs as opposed to providing generic services; and *patient-based care* emphasizes processes at the individual level as opposed to the unit or department level. Therefore, just because an organization collects patient perception data does not mean it is delivering PCC. Distinguishing among these similar concepts can be important as organizations progress towards patient-centeredness in their care. Additionally, the bulk of literature on PCC is still shaped by professional perspectives on what PCC should entail [20, 21], organizational models [22, 23] or health system recommendations [2, 9, 24]. Research on patients' perspectives on PCC is rare and often guided by, or conducted in conjunction with different professional or disciplinary perspectives [25–27]. This places emphasis on the need to incorporate patient's perspectives and their experience of PCC in evidence generation, and in ensuring the successful implementation of PCC [28–31].

In sub-Saharan Africa, improvements of care for people living with HIV and the rising incidence of chronic illnesses necessitates care that takes into account the psychosocial aspects of health and illness, and advocates for relationships that promote shared decision making between patients and their health workers. This in turn has led to a global increase in interest regarding PCC, especially at primary level [32]. This is also demonstrated by a wide range of community-based interventions including reforms in the care of HIV patients [33], supporting adherence to treatment and home based care [15, 34, 35], and encouraging more patient autonomy. Although these are not purely PCC interventions, they have aspects of patient-centeredness incorporated within their implementation. Similar interventions have also been carried out in Uganda [36]. In 2015, the Ugandan Ministry of health included PCC as one of the objectives of the health sector development plan and quality improvement strategy [37, 38]. However, its conceptualization is varied, implementation remains largely unmonitored and its impact on Uganda's health consumers and providers remains unmeasured [39].

Since primary healthcare facilities are the first point of care, this article aims to provide evidence of patients' perceptions of their experience of care and patient-centeredness at primary health facilities in Uganda; explore how patients perceptions varied according to the type of care they received (routine care, maternal and child health care and specialized care for specific health problems); and document if there are any key differences in the perception of care given by patients attending public versus private facilities. Using Uganda as a case, we make an empirical contribution to research, and the practice of PCC in similar contexts.

## Conceptual framework

The exploration of patients' perceptions on healthcare and patient-centeredness was based on a review of existing literature and tools used to measure patient perspectives of primary healthcare. Our conceptual framework was designed considering these three main areas of exploration: (1) to understand the patients' health needs, preferences and expectations, and factors

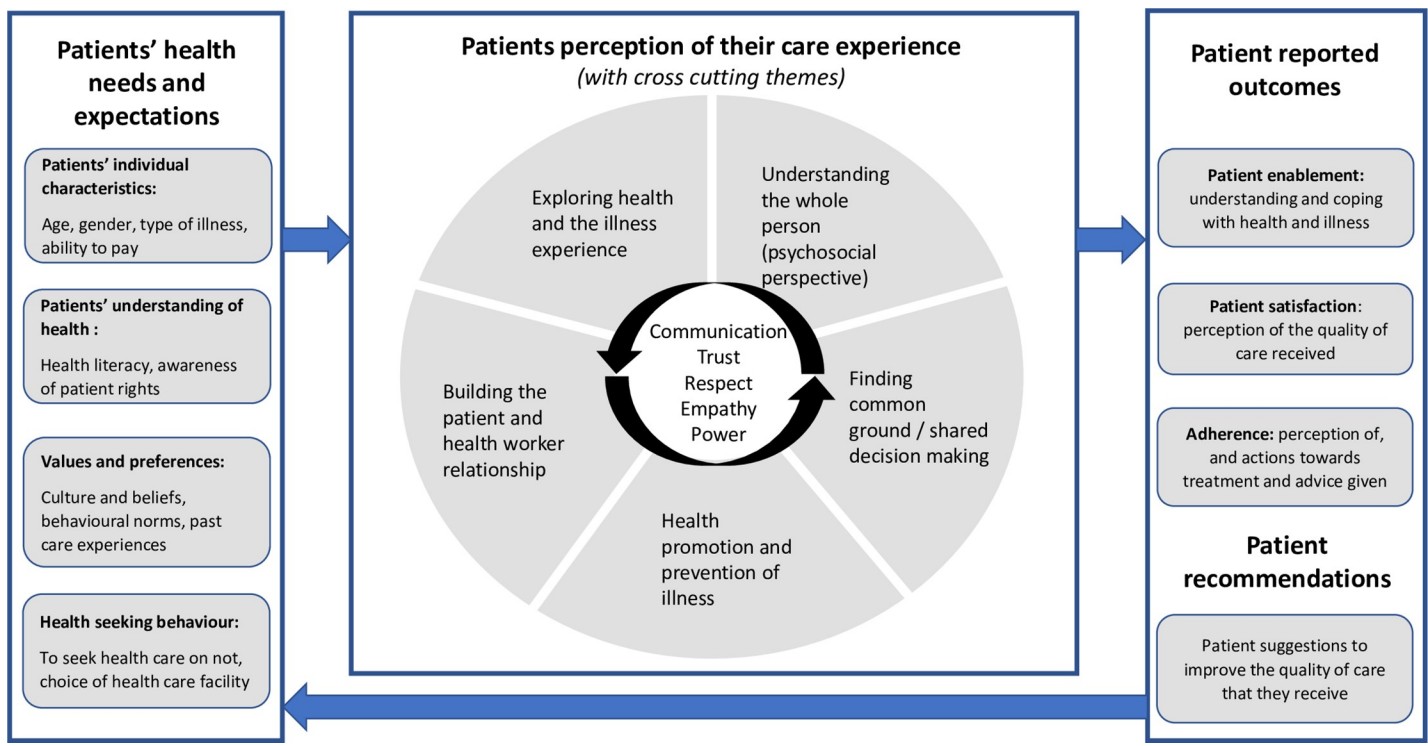

**Fig 1. Conceptual framework for exploring patients' perceptions of patient-centered care at primary health care level in Uganda.** A conceptual framework showing the main areas of exploration during data analysis including: (1) to understand the patients' health needs, preferences and expectations, and factors that influence their health-seeking behavior, (2) describe patients' perceptions of their care experience and (3) articulate patient-reported outcomes and their recommendations of how to improve the quality of care that they receive.

that influence their health-seeking behavior, (2) describe patients' perceptions of their care experience and (3) articulate patient-reported outcomes and their recommendations of how to improve the quality of care that they receive as shown in Fig 1.

The first part of the framework is about understanding patients' health needs and expectations. In detail, this involves looking at patients' individual characteristics (their understanding of health and care provision, awareness of their rights as patients); their values and preferences (which could be shaped by past experience, culture and beliefs); and how this influences their decision on whether and where to seek health care.

The second part of the framework describes patients' experience of PCC, related to five dimensions of PCC derived from the Stewart *et al*. and the Mead and Bower PCC models [40, 41]. These include:

1. Exploring the health, disease and illness experience: unique perceptions and experience of health, influence of patients' history on preferences, and the illness experience (feelings, ideas, effects on function and expectations)

2. Understanding the whole person: for patients this means going beyond physical illness to explore psychosocial aspects of health and illness, proximal context (family, social support) and distal context (culture, community). For health workers, it is looking at the factors that influence their practice including training, experience, mentorship, attitudes, motivation and the work environment

3. Finding common ground between the doctor and the patient: how problems are identified and prioritized, making decisions and setting goals of treatment and management, the perception of roles of patients and health workers

4. Prevention and health promotion: sources of health information, messages and follow up on treatment and management goals after health facility visits, effects on work and daily life activities, recognition of complications and when to seek health care

5. Building the patient and physician relationship: aspects that influence communication and interaction including trust, compassion, empathy, self-awareness and mindfulness

It is important to note that patients' experience of care may be influenced by contextual factors in addition to the interaction with the health worker. In this article we consider some of contextual factors like proximity of the facility to the patients' home, flexibility of opening and closing hours, waiting time, and ability to contact health workers if the facility is closed.

Thirdly, we wanted to measure how patients' experience of PCC contributed to patient-reported outcomes. Patient satisfaction–the extent to which patients are happy with their healthcare; and patient enablement–the extent to which a patient is capable of understanding and coping with his or her health issues, were chosen as our main outcomes of interest [42]. In addition, we chose to include patient's recommendations about how to improve care. However, we did not include patient adherence as a reported patient outcome due to the expected diversity of responses depending on the type of illness, whether the patient had to buy additional drugs, support at home and other factors that could not be comparable across patients or facilities.

## Methods

### Study design

This is a mixed methods cross-sectional study using patient exit survey questionnaires, semi-structured Interviews (SSIs), Focus Group Discussions (FGDs) and feedback meetings.

### Study location

This study was conducted between late 2017 and 2018 in the eastern Ugandan districts of Iganga and Mayuge, within the Iganga Mayuge Health and Demographic Surveillance Site (IMHDSS). It has a population of about 67,000 people in about 13,000 households. The IMHDSS is located on the boundary between the districts of Iganga and Mayuge, about 115 km from the capital Kampala. The area is predominantly rural with only about 10% living in a peri-urban environment. The majority of people are of Busoga culture and speak of local language of Lusoga. The Ugandan health system is organized into six levels of health care: level I comprises of village health teams (VHTs) and community medicine distributors; Health Centre II (HC II) led by an enrolled nurse; Health Centre III (HC III) led by a senior clinical officer; Health Centre IV (HC IV) and/or district hospital led by a senior medical officer; regional referral hospitals, and finally the National Referral and Teaching Hospital. Our study was conducted in the IMHDSS catchment area that has one district hospital, four government HC IIIs, three non-government HC IIIs, five government HC IIs and three Non-Governmental Organization (NGO) HC IIs.

We selected level III facilities to enable the analysis of perceptions from patients visiting the facility for different forms of curative care. We specifically focused on people who came "once-off" at the clinic (routine care); women/parents coming as part of a pregnancy/child follow-up with planned re-visits whereby interpersonal relations with staff are bound to develop

(maternal and child health care); and patients with chronic illnesses coming for follow-up in specialized clinics where repeated visits and external evaluations result in different relationships to the health practitioners, health system and different expectations in terms of quality of care (specialized clinics).

A HC III has, on average, about 18 staff, led by a senior clinical officer, with a general outpatient clinic, a maternity ward and a laboratory. We selected five HC III facilities according to the following criteria: inclusion of both governmental as well as private health facilities in both urban and rural settings; different demographic and epidemiological characteristics; different experiences with community strategy and PCC approaches, if any. We selected three public HC IIIs and three private-for-profit HC IIIs (see Table 2). Private health facilities are mostly located in semi-urban areas and have more clinical staff. Despite these differences, utilization rates, calculated as number of contacts per inhabitant (based on 2014 census) per year, are roughly similar in public and private facilities (ranging from 0.30 to 0.42 contacts per inhabitant per year) with the exception of facility 3 (utilization rate of 0.56). The utilization of routine all-round care services is many times higher than the utilization of maternal and child health care, and of more specialized care. Notable is that facility 6 is rather atypical: it has a substantially lower catchment population with fewer patients (in absolute numbers) using its services than is the case in the other five facilities, while focusing on care for pregnant women (see Table 2 in the results section). We decided therefore to exclude it from our analysis. Further detailed descriptions of the stakeholders involved in the provision of PCC at primary health care level in Uganda can be found in a paper by Waweru *et al.* [39].

## Data collection

**Development of tools to measure patient perceptions.** Using the framework described, we developed three data collection tools: patient exit questionnaires, semi-structured interview guides and focus group discussion topic guides (see S1 Appendix). Validated tools and questions were also added [40, 43] and a tool used to measure quality of care at health centre level in Zimbabwe [44]. Further detail on how literature and existing validated tools contributed to the design of tools to measure PCC in Uganda can be found in Supplementary file S1 Table—a table that compiles a detailed list of the dimensions (components of PCC), and the contribution (adopted questions) of each validated instrument to each dimension.

The patient exit questionnaire included questions about why they chose to visit the facility, who they interacted with, and how long they had to wait at each health point. Questions about the patient's perception of the care they received that day included a Likert scale score structured according to the 5 dimensions of PCC (exploring perceptions on health and the illness experience, understanding the whole person, finding common ground, enhancing the patient doctor relationship, and health promotion). Semi-structured interviews (SSIs) were conducted to solicit patients' perceptions on health-seeking behavior, who they thought was responsible for their health, their experiences at health facilities, their relationships with health workers and VHTs, their membership in support groups, their awareness of their rights and responsibilities as a patient, as well as how all these factors contribute to their perceptions of the quality of primary health care available to them (see sample questions in supporting file S1 Appendix). A follow-up Focus Group Discussion (FGD) was held with patients from each facility–the same as those who had participated in the SSIs–to validate and clarify the key messages (see S1 Appendix). As explained above, we did not collect data on adherence as the measurement techniques vary for different services and illnesses.

**Sample calculation, training of field assistants and piloting.** For the quantitative patient survey, we estimated that a sample size of 240 patients would allow us to detect differences in

average patient perceptions on their experience of PCC using frequencies between groups (public versus private) and services received (routine care, maternal and child health care or specialized care for patients with chronic illness) [45]. 60 patients were recruited per facility (20 patients receiving routine care, 20 receiving care at specialized clinics and 20 receiving curative care at the maternal and child health clinic). Our final sample of 300 patients, after the exclusion of patient responses from facility 6 was sufficient to describe the difference in mean scores for each dimension, and detect a mean difference of 0.3, with a significance level of 5% ($p < 0.05$), at 90% power after accounting for 30% attrition, as outlined in a review on self-management interventions for people living with HIV/AIDS in Africa by Aantjes *et al.* [46].

Four field assistants were selected based on their experience with both quantitative and qualitative research, three of them had also worked with the IMHDSS teams previously. They were trained for 2 weeks on the concept of PCC, how to administer the patient exit questionnaires and how to moderate a focus group discussion. At the end of the training period the field assistants (under the supervision of the Principal Investigator (PI)) piloted the consent forms and tools with patients.

From the pilot testing, we edited some of the questions, for example, the questions on how many nurses, clinical officers or lab technicians was generalized to 'how many health workers did you interact with today' as we discovered patients could not distinguish cadres of staff (including ourselves). Some words like patient rights and responsibilities (*idemberio*) were not easily understood and sometimes had to be explained from the starting point of a child's right to be fed, protected etc.; and the mother's responsibilities towards the child. Additionally, we also carried a summary of the patient rights charter to list the rights where the respondent did not understand completely. Empathy was also a concept that was understood as sympathy or taking pity and we had to train the field assistants to ask the question in order for the patient to understand it as 'the health worker putting themselves in your position (wearing your shoes) and sharing your feelings'.

**Exit questionnaires.** Of the patients visiting the five HC IIIs, 300 patients were recruited for exit surveys and 31 patients were purposively recruited for semi-structured interviews which were conducted at their homes. At each facility, the principal investigator (PI; first author EW) or a field assistant provided study information during the morning health talks and gained initial consent from patients interested to participate. Only patients receiving curative care at the out-patient department of the five HCs were included in the study. Furthermore, an effort was made to ensure that interviewees were representative of the three categories of patients receiving (1) routine care, (2) curative maternal and child health care, or (3) attending a specialized clinic (people living with diabetes or HIV/AIDS). If the patient had provided initial consent, EW would request to sit in during their consultation with the health worker. After patients received curative care, a sequential sample was taken where every 5th patient for the routine care department; every 2nd patient at the MCH or specialized clinic were recruited for an exit questionnaire interview. Only two patients refused to be interviewed citing lack of time and they were replaced. An interviewer-administered questionnaire was filled in for each selected patient / caregiver (in either the local language of Lusoga or English) (see S1 Appendix). At each facility 60 exit interviews were conducted.

**Interviews and focus group discussions.** After the questionnaire, a request was made to visit the patient at their home or a convenient location and date for them, for an SSI, including 2 patients from each of the 3 service areas. We also tried to keep a balance between male and female participants. In total, 31 SSIs were conducted (one facility had an extra patient interviewed).

A follow-up FGD was held with patients from each facility–the same as those who had participated in the SSIs–to validate and clarify the key messages. In total five FGDs were

**Table 1.  Summary of data collected from patients at primary health care facilities in Uganda.**

|  |  | Patient exit interviews | Semi-structured interviews | Focus group discussions (Number of patients in brackets: m-male, f-female) | Feedback talks with patients |
|---|---|---|---|---|---|
| **Public primary health care facilities** | Facility 1 | 60 | 7 | 1 (4m, 3f) | 1 |
|  | Facility 2 | 60 | 6 | 1 (2m, 4f) | 1 |
|  | Facility 3 | 60 | 6 | 1 (2m, 3f) | 1 |
| **Private primary health care facilities** | Facility 4 | 60 | 6 | 1 (3m, 3f) | 1 |
|  | Facility 5 | 60 | 6 | 1 (4m, 2f) | 1 |
| **Total** |  | **300** | **31** | **5 (15m, 15f)** | **5** |

conducted with 30 patients (5–7 patients in each FGD). In addition, 5 feedback talks were organized per facility with all patients present at the facility during morning health talks on the day of the scheduled feedback meeting (see Table 1). Data was collected between February and August 2018.

**Table 2.  Characteristics of selected facilities and the number of patients receiving care at primary healthcare facilities in 2017.**

|  |  | Facility 1 | Facility 2 | Facility 3 | Facility 4 | Facility 5 | Facility 6 |
|---|---|---|---|---|---|---|---|
| **Type of primary health care facility** |  | Rural, Iganga district | Semi-Urban, Iganga district | Rural, Mayuge district | Semi-Urban, Iganga district | Semi-Urban, Iganga district | Rural, Iganga district |
|  |  | Public HC III | Public HC III | Public HC III | Private HC III | Private HC III (faith-based) | Private HC III |
| **Catchment population of corresponding sub-county according to the 2014 census** |  | 42772 | 50478 | 30896 | 55263 | 29000 | 3583 |
| **2017 utilization (number of patients)** |  | 13328 | 16530 | 17290 | 20051 | 11827 | 1516 |
| **Number of patients per year per type of service** | Routine care | 10598 | 14911 | 11769 | * | 10203 | 549 |
|  | Maternal and child health care | 1540 | 1211 | 2129 | * | 697 | 958 |
|  | Specialized care | 1190 | 408 | 3392 | * | 927 | 9 |
| **Utilization rates: number of contacts per inhabitant per year** |  | 0.31 | 0.33 | 0.56 | 0.36 | 0.41 | 0.42 |
| **Number of clinical Staff including clinical officers, nurses and laboratory technicians** |  | 12 | 9 | 12 | 31 | 22 | 4 |
| **Services offered in addition to routine Outpatient Care, Maternal and Child Health Care, laboratory testing services (which are offered by all)** |  | Specialized care for patients with HIV/AIDS and diabetes | Specialized care for patients with HIV/AIDS | Specialized care for patients with HIV/AIDS and diabetes | Specialized care for patients with HIV/AIDS and diabetes, Inpatient care, dental care, ultra-sound and minor surgeries | Inpatient care and minor surgeries | Specialized in attending to pregnant women and young children |

*for facility 4, we obtained an overall annual utilization rate, but have not the data to disaggregate utilization per type of service used

## Data analysis

Data analysis was conducted according to the conceptual framework i.e. categorized into three major areas of exploration: (1) understand the patients' health needs, preferences and expectations, and factors that influence their health seeking behavior, (2) describe patients perceptions of their care experience, and (3) articulate patient reported outcomes and their recommendations of how to improve the quality of care that they receive (see Fig 2).

Data from the survey questionnaire were entered into excel worksheets at the end of each day of data collection. Random checks were conducted by the PI to ensure completeness of data at the field. The compiled data was then imported to STATA version 14 where it was cleaned and data from one of the facilities was excluded as explained above (see data collection section). Although the ordinal variables have a rank order but cannot be conceived as having an underlying measurable standard (e.g. the interval difference between a strongly agree and agree response is not standardized), data from patient reported measures have been coded in this way to provide feedback that managers and policy makers can easily understand. For each question, strongly agree was recoded as 5 and strongly disagree as 1 so that higher values could reflect more patient-centeredness. Subsequently, for each of the PCC dimensions above, patients' responses were then summed and normalized to fit on a scale of 1–100 i.e. (the sum of the scores / (number of questions*5))*100. This normalized score enables a comparison of patient experiences: across the 5 PCC dimensions, depending on the type of facility (public or private PHC), and according to the service that patients received (routine care, maternal and child health care and specialized clinic care for HIV/AIDS or Diabetes). The results were summarized in terms of the frequency of patients who gave positive or negative perceptions of

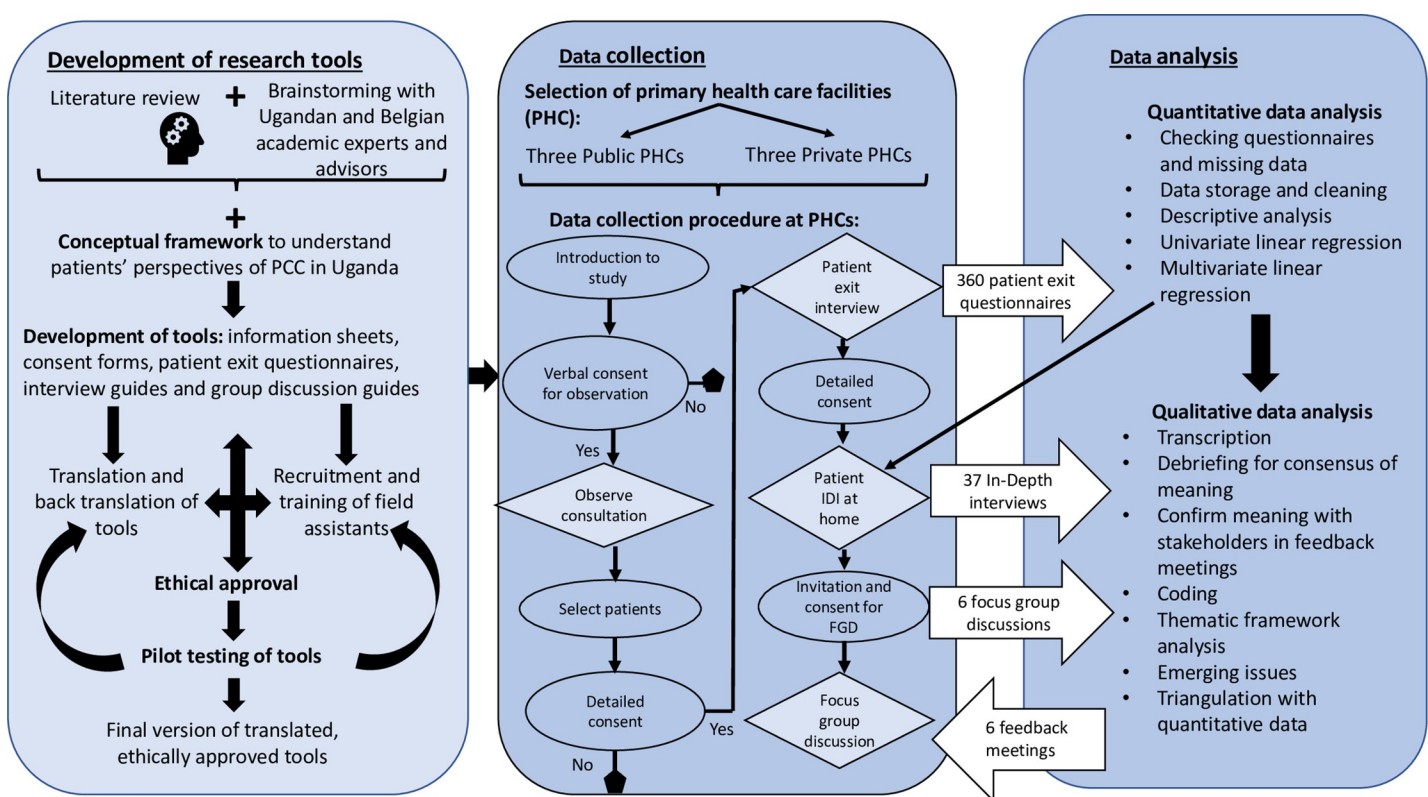

**Fig 2. Procedures for data collection and analysis plan for study on patient perceptions of PCC in rural eastern Uganda.**

PCC in public and private facilities, and normalized scores were calculated. For this article, chi-squared tests have been used to determine whether there are significant differences reported between patients attending public and private facilities (with a null hypothesis that there was no difference in patient-centeredness between public and private facilities). Within each category (public/private facilities) further descriptive analysis was conducted to describe differences in the perception of PCC in patients receiving routine care, maternal and child health care or specialized care. Determinants of the outcomes of interest (patient satisfaction and patient enablement) were investigated by performing univariate and multivariate regressions with patient perceptions of PCC, interaction effects, facility type, type of service received, age, gender and literacy as independent variables. Data on adherence was not presented because responses were too varied as they depended on the type of illness, whether the patient had to buy additional drugs, support at home and other factors that could not be comparable across patients or facilities.

Qualitative data was converted into digital text format for analysis. Audio recordings of SSIs and FGDs were transcribed, and notes from feedback sessions were typed. Any text in Lusoga was translated into English. The written text was imported into NVivo 11, a qualitative data analysis software, for organization and qualitative thematic framework analysis as described by Gale *et al.* [47]. Thematic framework analysis of SSIs and FGDs was begun in the field by the first author. Following the first phase of data collection using patient exit questionnaires, qualitative questions were developed by the research team to clarify patient responses in the patient exit survey interviews. Emerging themes were added on while conducting the SSIs, and during de-brief meetings with translators. The themes were further explored and the information validated during follow-up FGDs and feedback meetings with patients. Post data collection, the research team developed a coding framework for use in NVivo 11 guided by the conceptual framework for understanding patients' perception of PCC, and collected data. Data was coded according to the framework in NVivo 11 and organized into framework matrices or charts. Interpretation of data was conducted by the research team including the PI, research assistants who spoke the local language and supervisors. The interpretation of data was also validated through feedback meetings with patients, health workers and district health managers.

### Ethics statement

Ethical approva**l** for this study was obtained from the Institute of Tropical Medicine PhD committee (IRB/AB/ac/081), the Institute of Tropical Medicine Institutional Ethics Review Board (1166/17); University of Antwerp Ethics Review Board (17/24/278); the Makerere University School of Public Health Institutional Review Board (500); and the Ugandan National Council for Science and Technology. Prior to participation, written informed consent was obtained from stakeholders at national, district and facility level. In cases where patients could not write or provide a signature, verbal informed consent was obtained from patients and recorded in the study's copy of the informed consent form.

### Results

This section will present the results in three parts as described by the conceptual framework, i.e. (1) patient needs and expectations, (2) patients' perception of their care experience, and 3) the relation of patient experience to patient reported outcomes. We begin by describing the characteristics of the primary health care facilities, and the demographic characteristics of the 300 patients in public and private facilities.

Table 2 presents the principal characteristics of the six facilities investigated, including the services on offer. Efforts were made to ensure a balance between public and private facilities, an inclusion of facilities in both rural and urban settings with comparable facility utilization. This led to the exclusion of facility 6 in subsequent analysis comparisons (see selection criteria section).

Table 3 shows that most of the patients interviewed were between the age of 25 and 44 years (52.33%), female (79.00%) and could not read English (63.00%). Literacy was measured with the ability to read English because health information and information on patient rights was presented as visual aids written in English and posted on the facility walls. We notice a number of differences in the patient population groups attending public PHCs and the ones going to the private PHCs: the latter appear to be younger and more literate compared to the patients used the public PHCs. This is presented in Table 3 below.

## Patients' needs and expectations

The first part of the conceptual framework conveys that patients' needs and expectations can be shaped by the individual characteristics, values, preferences, patients' understanding of health and health service provision. When patients were asked how they feel about their health and the health of their families, they gave mostly positive responses. Good health was related to a clean home and the availability of clean food and water. As expected, most patients also reported seeking healthcare only when a member of the family was feeling unwell. While the caregivers (mostly mothers) were the first to notice when a member of the family was ill, the decision to seek healthcare, and where to go was mostly made by the male heads of the households. Specific to mothers of young children was also the importance of taking children for immunization. When asked about sources of health information most patients reported getting their information from health workers, the radio, local council members (information on sanitation) and Village Health Team (VHT) members. The responsibility for health was perceived to be more the responsibility of the health worker than of the patient.

**Table 3. Demographic characteristics of interviewees involved in research exploring PCC in rural eastern Uganda.**

|  | Public PHCs | Private PHCs | Total |
|---|---|---|---|
| Number of facilities | 3 | 2 | 5 |
| Number of patients | n = 180 | n = 120 | N = 300 |
| **Service received at the facility** |  |  |  |
| Routine care | 60 (33.33%) | 40 (33.33%) | 100 (33.33%) |
| Maternal and child health care | 60 (33.33%) | 40 (33.33%) | 100 (33.33%) |
| Specialized care (Diabetes or HIV/AIDS clinic) | 60 (33.33%) | 40 (33.33%) | 100 (33.33%) |
| **Age** |  |  |  |
| 16–24 years | 55 (30.56%) | 25 (20.83%) | 80 (26.67%) |
| 25–44 years | 76 (42.22%) | 81 (67.50%) | 157 (52.33%) |
| Above 45 years | 49 (27.22%) | 14 (11.67%) | 63 (21.00%) |
| **Gender** |  |  |  |
| Male | 35 (19.44%) | 28 (23.33%) | 63 (21.00%) |
| Female | 145 (80.56%) | 92 (76.67%) | 237 (79.00%) |
| **Literacy*** (can read a letter written in English) |  |  |  |
| No | 137 (76.11%) | 52 (43.33%) | 189 (63.00%) |
| Yes | 43 (23.89%) | 68 (56.67%) | 111 (37.00) |

Abbreviation PHC stands for primary health care facility.

## Box 1. Confirmatory quotes on patient expressions about their health needs and expectations

| Topic | Exemplifying quote |
|---|---|
| Patient perception of health | *"I think the health of my family is good, I keep the house clean and feed the children well, I make sure there is also clean water for drinking, and they go to school"* (MCH patient, Female, 31yrs, facility 1-public) <br> *"I am the main care giver now, their mother passed away last year, so I have to make sure I take my medication, that the children eat well and go to school, I also talk to them when they are sad from time to time* (special clinic for ART patient, Male, 27yrs, facility 4-private) |
| Patient expectations of health care and health seeking behavior | *"It depends on how serious the disease is, first we give Panadol. If it doesn't resolve we go to the drug shop nearby, but when the condition is serious we just go to the health facility"* (MCH patient, female, 29yrs, facility 3-public) <br> *"When someone in my house gets sick, I take them to hospital, unless it is at night, I give them Panadol as first aid then go to hospital in the morning"* (MCH patient, female, 42yrs, facility 5-private) |
| Sources of health information | *"There are those people* [VHT members] *who go around telling mothers about immunization, then there are those who go around to make sure we have clean toilets* [local council members]*"* (MCH patient, female, 38yrs, facility 4-private) <br> *"Sometimes they also make announcements on the radio about keeping toilets clean, when there are free services at the health centre and so on"* (routine care patient, male, 29yrs, facility 3-public) |

There were no key differences in the responses of patients receiving care at public and private facilities, regarding their perceptions of health needs and expectations, exemplifying quotes are presented in Box 1.

There were however some important differences in what patients expected while receiving care at public and private facilities. Patients using public facilities chose to do so because the public primary care facility is near to their homes (see Table 4: 78.33% in public PHCs versus 23.33% in private PHCs) and its services are free of charge. On the other hand, patients attending private facilities said to do so because of the perception of better quality of care (84.17% in private PHCs versus 21.67% in public PHCs). Patients attending public facilities were also less positive about the facility being open at night or during holidays. In contrast, private facilities appear to be much more responsive through their offer of flexible opening hours, more options to contact health workers, and seeing the patients on the same day even when the facility is closed. With regards to referrals, above 60% of patients attending both public and private primary health care facilities reported positive experiences with getting referral notes to see a doctor at the district hospital. However, less than half (47.67%) reported discussing the outcome of the referral with their primary health worker during an occasional subsequent visit. Lastly, while it took most patients, whatever the type of facility used, less than an hour to get from their home to the facility, patients in public facilities spent more time at the facility with slightly below half (42.78%) reporting that they spent more than two hours waiting–excluding the time spent with the health worker and time taken for the interview. In contrast, patients receiving care in private facilities spent less time at the health care facility (see Table 4).

Follow-up SSIs and FGDs confirmed that most patients valued the close proximity, fast service and availability of medication at drug shops especially at night. VHTs were also reported as

**Table 4. A comparison of patient expectations of care in public and private facilities.**

| | Total Public PHC | Total Private PHC | Total number of patients |
|---|---|---|---|
| | n = 180 | n = 120 | N = 300 |
| | Number of patients (percentage of patients in this category) | Number of patients (percentage of patients in this category) | Number of patients (percentage of patients in total) |
| **Why did you choose to come to this facility (public or private PHC)? (patients selected all options that applied therefore column total is not equal to 100%)** | | | |
| Nearest to my home | 141 (78.33%) | 28 (23.33%) | 169 (56.33%) |
| It is free | 100 (55.56%) | <5 (3.33%) | 104 (34.67%) |
| The quality of services is good | 39 (21.67%) | 101 (84.17%) | 140 (46.67%) |
| I like the health worker | 11 (6.11%) | 30 (25.00%) | 41 (13.67%) |
| I was referred here | <5 (2.22%) | <5 (3.33%) | 8 (%) |
| **Patients' perception of facility opening times and availability of health workers** | **Total Public PHC** | **Total Private PHC** | **Total number of patients** |
| | **n = 180** | **n = 120** | **N = 300** |
| **The facility is open on:** | | | |
| Weekdays | 160 (88.89%) | 117 (97.50%) | 277 (92.33%) |
| Weekends | 132 (73.33%) | 117 (97.50%) | 249 (83.00%) |
| At night | 43 (23.89%) | 113 (94.17%) | 156 (52.00%) |
| Public holidays | 42 (23.33%) | 113 (94.17%) | 155 (51.67%) |
| **If the facility is open:** | | | |
| Someone will see me the same day | 180 (100.00%) | 118 (98.33%) | 298 (99.33%) |
| I will see the same health worker | <5 (0.56%) | <5 (2.50%) | <5 (1.33%) |
| I can request a specific health worker | 62 (34.44%) | 45 (37.50%) | 107 (35.67%) |
| **If the facility is closed:** | | | |
| I will be seen on the same day | 19 (10.56%) | 50 (41.67%) | 69 (23.00%) |
| There is a number I can call | 14 (7.78%) | 20 (16.67%) | 34 (11.33%) |
| I can call a specific health worker on the phone | 21 (11.67%) | 17 (14.17%) | 38 (12.67%) |
| **Coordination of care** | **Total Public PHC** | **Total Private PHC** | **Total number of patients** |
| **Patient perceptions on referrals to see doctor at the district hospital** | **n = 180** | **n = 120** | **N = 300** |
| When I see a doctor at the district hospital, the health worker has to approve | 125 (69.44%) | 80 (66.67%) | 205 (68.33%) |
| When I see a doctor at the district hospital, the health worker writes a note | 126 (70.00%) | 80 (66.67%) | 206 (68.67%) |
| When I see a doctor at the district hospital, I discuss the results with the PHC health worker | 82 (45.56%) | 61 (50.83%) | 143 (47.67%) |
| **Time** | **Total Public PHC** | **Total Private PHC** | **Total number of patients** |
| | **n = 180** | **n = 120** | **N = 300** |
| **How long did it take you to get to the facility** | | | |
| 30 minutes or less | 116 (64.44%) | 106 (88.33%) | 222 (74.00%) |
| 31–60 minutes | 47 (26.11%) | 13 (10.83%) | 60 (20.00%) |
| 61–120 minutes | 16 (8.89%) | <5 (0.83%) | 17 (5.67%) |
| Above 120 minutes | <5 (0.56%) | <5 (0.00%) | <5 (0.33%) |
| **Total time spent waiting at the facility (excluding time with the health workers)** | | | |
| 30 minutes or less | 15 (8.33%) | 19 (15.83%) | 34 (11.33%) |
| 31–60 minutes | 25 (13.89%) | 37 (30.83%) | 62 (20.67%) |
| 61–120 minutes | 63 (35.00%) | 57 (47.50%) | 120 (40.00%) |
| Above 120 minutes | 77 (42.78%) | 7 (5.83%) | 84 (28.00%) |

## Box 2. Exemplifying quotes on patient expectations, preferences and choice of facilities

| Topic | Exemplifying quotes |
|---|---|
| Patient needs and expectations before the visit | *"We go to the facility to get care and medication, sometimes you'll get what you need, sometimes you will not. It depends on whether the health worker is there, if they can do the tests and if there are medicines in the store; if you are unlucky, you have to go and get tested or buy medication somewhere outside the facility. . .*" (routine care patient, male, 39yrs, facility 2-public) <br> "Here (at the facility) they have everything, it's just about if you can pay for it" (routine care patient, female, 29yrs, facility 5-private) |
| Patient preferences | *"I like the health workers here, they take good care of us and so if I have to take a boda boda to reach here, it is okay"* (special clinic for diabetic care patient, male, 49yrs, facility 1-public) <br> *"Here they have a lab where you can get all the tests done and you can get all the types of drugs for the sickness"* (MCH patient, female, 29yrs, facility 5-private) <br> *"When I go to a doctor or the VHT, I don't want them to tell others–that one is suffering from Syphilis, or AIDS–so I would rather pay a little money and have peace"* (special clinic for ART patient, male, 39yrs, facility 4-private) |

useful in interpreting health information about preventive care (e.g. immunization), but were not consulted when someone was ill or had social problems (see Box 2 for exemplifying quotes).

### Patients' perception of their care experience

This section deals with patients' perceptions of patient-centeredness and their care experience (see Fig 1). To begin, Fig 3 is a Likert plot that shows the responses to individual questions related to the dimensions of PCC.

The bars indicate the percentage of patients who responded strongly agree, agree, neutral, disagree or strongly disagree to each question/statement. In order to facilitate interpretation, neutral, disagree and strongly disagree were plotted as negative responses (on the left side); while agree and strongly agree were plotted as positive responses (on the right side) as shown in Fig 3. More detail on the individual questions asked in the exit questionnaire can be found in supplementary file S2 Table.

Table 5 shows that overall, patients perceived services to be slightly more patient-centered in private facilities (60.6) than in public facilities (57.54). High scores (above 70) were given in exploring health and the illness experience. These scores remained similar for patients receiving different types of services in both public and private PHCs. With regards to understanding the whole person, overall scores were slightly below the half way score of 50, with public facilities having slightly higher scores than private facilities. In public facilities, patients receiving specialized care gave higher scores (56.2) than those receiving routine care (47.2) or maternal and child health care (46.3). In private facilities, patients receiving specialized care also gave the highest scores (54.8). Looking at shared decision making, scores for both public and private facilities were around the half way mark. However, it is clear that patients receiving maternal and child health care gave the lowest scores (44.5). Health promotion was better perceived in private (60.7) than in public facilities (54.3). Lastly, patients' perceptions of their relationships with health workers were similar across types of facilities and according to the services they received (ranging from 62.1 to 64.9).

Linear regression was conducted to estimate the differences in normalized scores of PCC across type of PHC and care provided (see data analysis section for description on normalized

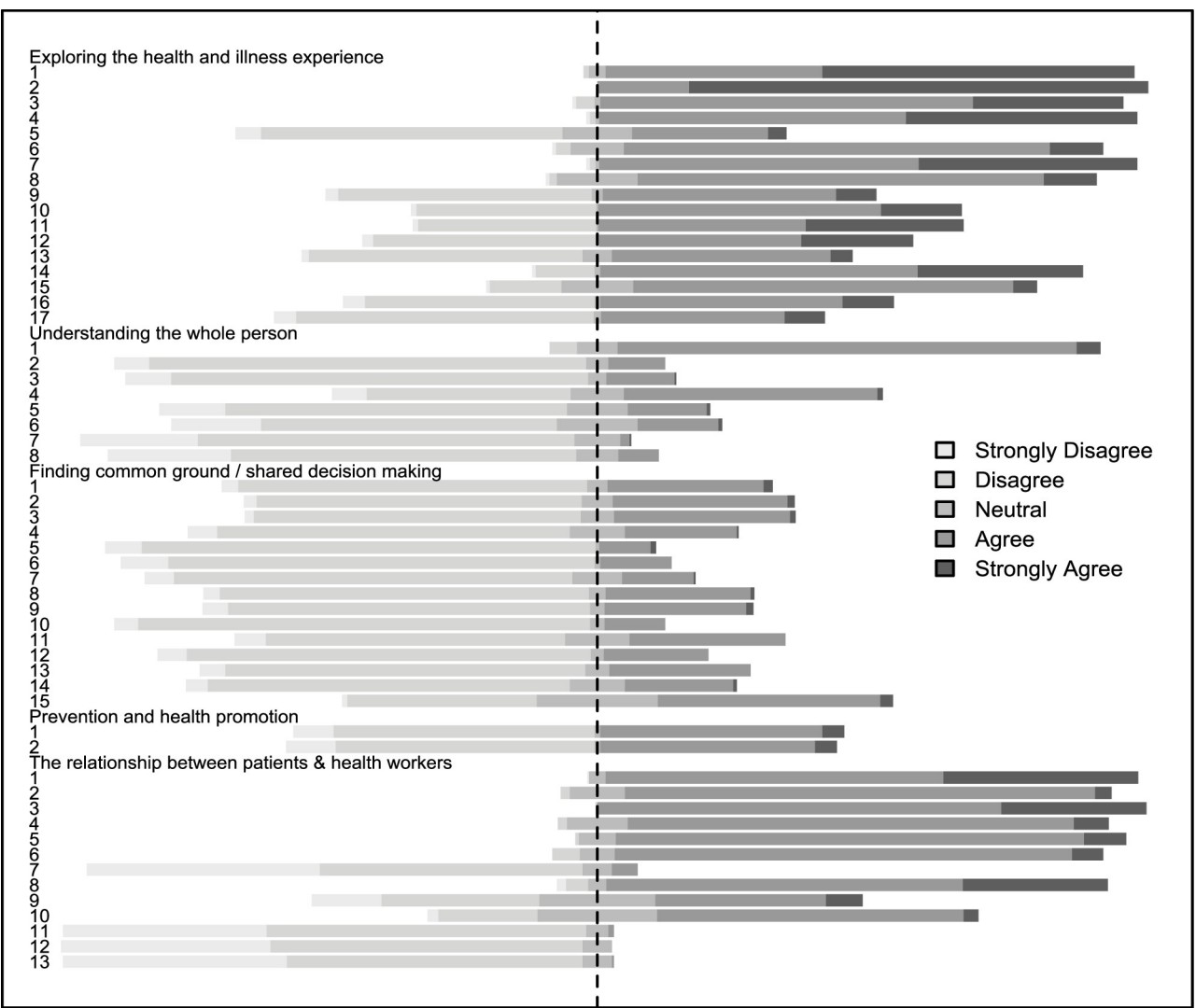

**Fig 3. Patients' perceptions of patient-centered care dimensions and patient reported outcome measures.** Fig 3 is a Likert plot that shows the responses to individual questions related to the dimensions of PCC. The bars indicate the percentage of patients who responded strongly agree, agree, neutral, disagree or strongly disagree to each question/statement. In order to facilitate interpretation, neutral, disagree and strongly disagree were plotted as negative responses (on the left side); while agree and strongly agree were plotted as positive responses (on the right side). More detail on the individual questions asked in the exit questionnaire can be found in supplementary file (see S2 Table).

scores). The results are displayed in Table 6 below. For PCC dimension 1—exploring health and the illness experience, there was a five-point significant difference with patients receiving care in private PHCs giving higher scores (P<0.001). For this first dimension, among the patients using the public facilities, those receiving routine care reported the highest scores followed by maternal child healthcare and lastly patients receiving specialized care as indicated by negative differences when comparing these services to routine care. This was however different to patients who visited private facilities where patients receiving maternal and child healthcare reported the highest PCC scores with a positive coefficient of 3.2 points when compared to routine care. For PCC dimension 2—understanding the whole person, there were (non-significant) differences in comparing specialized care to routine care for both public (9-point difference) and private facilities (10.5-point difference). For PCC dimension 3—

 

**Table 5. Descriptive results when patient responses are summed and normalized to fit in a PCC point score of 1–100.**

| Normalized score out of 100 (higher scores interpreted as more patient-centered) | RC | MCH | SC | Total Public PHC (Confidence interval) | RC | MCH | SC | Total Private PHC (Confidence interval) |
|---|---|---|---|---|---|---|---|---|
| | n = 60 | n = 60 | n = 60 | | n = 40 | n = 40 | n = 40 | |
| PCC dimension 1: Exploring health and the illness experience | 73.5 | 71.7 | 70.6 | **71.9 (28.2–97.6)** | 75.3 | 78.4 | 77.1 | **76.9 (61.1–98.8)** |
| PCC dimension 2: Understanding the whole person | 47.2 | 46.3 | 56.2 | **49.9 (27.5–90)** | 44.3 | 46.8 | 54.8 | **48.6 (25.0–87.5)** |
| PCC dimension 3: Finding common ground-decision making | 48.4 | 44.5 | 52.1 | **48.4 (25.3–80)** | 47.8 | 52.9 | 56.5 | **52.4 (28–84)** |
| PCC dimension 4: Health promotion | 57.5 | 51.8 | 53.5 | **54.3 (20–100)** | 61.8 | 54.8 | 65.5 | **60.7 (20–100)** |
| PCC dimension 5: Relationship between patient and health worker | 62.1 | 63.2 | 64.4 | **63.2 (46.2–76.9)** | 63.9 | 64.9 | 64.6 | **64.4 (52.3–78.5)** |
| **Total mean** | **57.74** | **55.5** | **59.36** | **57.5 (37.6–78.5)** | **58.62** | **59.56** | **63.7** | **60.6 (45.5–83.1)** |

RC: Routine care.

MCH: Maternal and child health care.

SC: Specialized care for people living with HIV/AIDS or Diabetes.

PHC: primary health care facility.

**Table 6. Testing the significance of differences in normalized PCC scores according to the type of facility patients visited, and the type of service they received, in rural eastern Uganda.**

| Difference of normalized scores between respondents attending public and private PHCs (with public as the reference category) | | | | |
|---|---|---|---|---|
| | Co-efficient (confidence interval) | | | |
| PCC dimension 1: Exploring health and the illness experience | **5.0 (2.9–7.1)** *** | | | |
| PCC dimension 2: Understanding the whole person | -1.3 (-3.8–1.2) | | | |
| PCC dimension 3: Finding common ground-decision making | **4.1 (1.3–6.8)** ** | | | |
| PCC dimension 4: Health promotion | **6.4 (1.1–11.6)** * | | | |
| PCC dimension 5: Relationship between patient and health worker | 1.2 (-0.0–2.5) | | | |
| | **Public PHCs** | | **Private PHCs** | |
| | Comparing MCH to Routine care | Comparing Specialized care to Routine care | Comparing MCH to Routine care | Comparing Specialized care to Routine care |
| | Coefficient (Confidence interval) | Coefficient (Confidence interval) | Coefficient (Confidence interval) | Coefficient (Confidence interval) |
| PCC dimension 1: Exploring health and the illness experience | -1.8 (-5.1–1.5) | -3.0 (-6.3–0.4) | 3.2 (-0.6–7.0) | 1.9 (-2.0–5.7) |
| PCC dimension 2: Understanding the whole person | -0.8 (-4.3–2.7) | 9 (5.5–12.5) | 2.6 (-1.8–7.0) | 10.5 (6.1–14.9) |
| PCC dimension 3: Finding common ground-decision making | -3.9 (-7.7 - -0.0) | 3.7 (-0.1–7.6) | 5.1 (-0.4–10.6) | 8.7 (3.2–14.2) |
| PCC dimension 4: Health promotion | -5.7 (-13.8–2.5) | -4.0 (-12.1–4.2) | -7.0 (-16.9–2.9) | 3.8 (-6.1–13.6) |
| PCC dimension 5: Relationship between patient and health worker | 1.0 (-0.8–2.9) | 2.2 (0.4–4.1) | 1.0(-1.5–3.5) | 0.7 (-1.8–3.2) |

MCH: Maternal and child health care.

PHC: Primary health care facility.

P value asterisk

* P<0.05

** P<0.01

*** P<0.001.

 

finding common ground and shared decision making, there was a positive 4-point significant difference in scores when comparing private facilities to public facilities (P<0.01).

Patients receiving maternal and child health care in public facilities gave, on average, lower scores than those receiving routine care, indicating that specialized care is associated with a lessened perception of being involved in decision-making. However, this association is reversed in private PHCs, where patients receiving specialized care gave higher scores than those receiving routine care. In terms of PCC dimension 4—health promotion, it seems that patients have better perceptions of care if they received routine care or specialized care in a private facility. Otherwise the coefficient differences are negative and more so for patients receiving maternal and child healthcare in private facilities. The overall comparison also shows that the 6.4 difference when comparing private facilities to public facilities is borderline significant (P<0.05). Results for the fifth dimension—relationship between patient and health worker, show that patients are likely to have slightly more positive perceptions of PCC if they were attended to in private facilities and if they received maternal and child care or specialized care compared to routine care. The differences are however slight and not found to be statistically significant (see Table 6 below).

Qualitative data was also collected to elaborate patient responses and gather information on factors that lead to perceived positive or negative experiences related to PCC. These are illustrated with exemplifying quotes in Box 3. In exploring *perceptions of health and illness*, most patients indicated that they perceived the health workers being responsible for, and most knowledgeable about the patients' health. A few patients also said God was responsible for healing. At home, the household head (in most cases a man) was responsible for making decisions about when and where to seek treatment. The women (mothers, daughters and grand-mothers) were responsible for ensuring homes were kept clean, children were fed, and that those who were ill followed the instructions given by the health worker or drug shop owner (the latter being visited when the illness was not perceived as very severe). In the Busoga culture, illness is part of everyday life and is even included as part of the morning greeting *(ebiku-luma)* which translates to *"how is your sickness"*. With the exception of immunization, healthcare is also only sought when one is ill. These cultural considerations are important in determining the patient preferences and health-seeking behavior. Depending on the perceived illness, most patients receiving care in public facilities preferred to give painkillers at home or speak to the attendant in a drug shop before going to the facility, especially if the episode of illness began at night (see Box 3 section on exploring the health and illness experience).

*Understanding the whole person* entails looking at the psychological, emotional and social aspects of healthcare. Most patients felt it was not the responsibility of the health worker to discuss emotional and psychological aspects of their health with many saying *"it is disturbing the health worker"*, and were also afraid of the health workers reaction to discussing personal issues. Consequently, they appropriated this role to family and friends. Patients did not discuss how their illness was affecting their day-to-day activities at home and were more attentive to instructions on medical treatment as opposed to diet or reducing stress. Patients receiving specialized care, and especially people living with HIV, received more encouragement to discuss how their illness was affecting their daily life. Some patients also gave examples of little things that health workers did to make them feel understood—like giving a clean t-shirt to a mother whose child had vomited on them, she greatly valued this action that made her feel seen and understood (see Box 3).

*Finding common ground* (or shared information and decision making) was perceived to be influenced by desirable characteristics of the patient and the health worker. Patients' felt that the communication with the health worker was enhanced when there was trust. This was expressed from pleasant past experiences, kind language, keeping patients' secrets and

## Box 3. Exemplifying quotes on patient experiences of care related to PCC

| | Experiences perceived as positive and promotive of PCC | Experiences perceived as negative and unresponsive to PCC |
|---|---|---|
| PCC dimension 1: Exploring health and the illness experience | **Responsibility for health** *"The health worker is responsible for my health. He is the one to do the test, know what is wrong and give me medication to make me go back to normal"*(routine care patient, male, 44yrs, facility 2-public) <br> *"I am the one who takes care of the home, I keep it clean, prepare the food and when my children are not feeling well, I am the first one to notice and tell my husband. . . he decides what we are going to do"* (MCH patient, female, 29yrs, facility 5-private) <br> **Health seeking behavior** *"It depends on how serious the disease is, first we give Panadol, if it doesn't resolve we go to the drug shop nearby, but when the condition is serious we just go to the health facility"* (MCH patient, female, 29yrs, facility 3-public) | **Difficulty in expressing illness and symptoms** <br> *"it was my first time to meet this health worker, there are many people waiting, I felt like he was rushed so there was no time for stories, i said my most pressing problems so that others can also be seen"* (routine care patient, female 31yrs, facility 2-public) |
| PCC dimension 2: Understanding the whole person | **Comfortable environment** *"I like this facility because I go to see the health worker in a room by myself so I am not afraid or ashamed, that other people will hear what I am saying. They also have a very positive attitude, they listen and they are kind, they cannot be rude. When you ask a question it is answered"* (routine care patient, male, 39yrs, facility 4-private) <br> **Interest in the effect of illness on daily life** *"it when the nurse told me that I was positive (living with HIV) but that I could still live a long life if I took medication and stayed healthy. She also counselled me on how to stay calm and helped me to share the news with my wife who was not happy at first but now we are working and living together"* (Special clinic for ART patient, Male, 33yrs, facility 3-public) <br> **The little things** *"I brought my son here and he was shaking and vomiting, he vomited on me. . . they took good care of my son and they also took care of me, they gave me a clean t-shirt and a nurse sat with me and counselled me, I was very touched, I pray for them every day"* (MCH caregiver with sick child, female 34yrs, facility 5-private) | **Emotional and psychological issues are not perceived as part of health or the responsibility of the health worker** <br> *"I would never talk about such issues with the health worker, those are my thoughts and my issues, if I cannot handle them I talk to my family or elder people, not the health workers, they are there to treat the sickness in my body"* (routine care patient, female 32yrs, facility 4-private) <br> *"Imagine if I went in to see the health worker and they asked me. . .what is the problem? And I answer I am having many thoughts or stress (laughs) they would just laugh and tell everybody at the facility"* (MCH patient, female, 46yrs, facility 1-public) <br> **Lack of privacy** *"When there are other patients around at the time when the health worker is talking to you, you cannot share intimate information, so you have to wait until the other people have been seen or you come on another day"* (routine care patient, female, 27yrs, facility 3-public) |
| PCC dimension 3: Finding common ground-decision making | **Health literacy** *"of us who are not very well informed, when we have questions, we are afraid to ask because we might disturb the musawo (health worker) but they write in the book and we go and ask the VHT what was written in the book"* (Special clinic for ART patient, male, 28yrs, facility 2-public) <br> **Information sharing** *"I always ask questions when I don't understand, because I am paying for this care, if I don't ask I will have to come back and pay again! The good thing is the health worker always answers and I understand what to do next"* (routine care patient, male, 34yrs, facility 4-private) <br> **Awareness of patient rights** <br> *"I read them on the wall every time I come to the facility so I know my rights and if I don't get them, I will ask to speak to the administrator"* (routine care patient, male, 33yrs, facility 4-private) | **Low expectations for involvement in care** <br> *"I am satisfied, I did not understand why they did the test (blood test) but I got medication, and I feel better"* (patient, male, 44yrs, facility 1-public) <br> *"I always bring the book with me but I don't know what is written in it, should I ask?"* (MCH patient, female, 20yrs, facility 2-public) <br> **Fear of asking questions** <br> *"when you ask questions or say I have a right to know, you just better get another health facility because the health workers will not want to deal with you, they say [the one with many questions has come]"* (routine care patient, female, 45yrs, facility 1-public) <br> **No avenues for conflict resolution** *"If I am not happy with a health worker or the service, there is no place or person I can go to, I keep it with myself or just shift to another facility"* (routine care patient, male, 41yrs, facility 3-public) |
| PCC dimension 4: Health promotion | **Information sources** *"here we get most of our information from people who go around with a public address system (confirmed as VHTs) sometimes they also make announcements over the radio for things like clean toilets or proper places we should keep waste"* (routine care patient, male, 39yrs, facility 4-private) <br> **Home visits** *"I was actually surprised a few weeks ago I was very sick and could not go to the facility to get my medication, but the social worker came all the way to my home to bring the medication, I was very happy and grateful, they should continue like that"* (special clinic for ART patient, male 29yrs, facility 4-private) <br> **Individual advice** *"after getting examined, the health workers always tells me if my pressure (blood pressure) is improving or bad, they emphasize that I need to eat well and exercise"* (special clinic for diabetes patient, female, 43yrs, facility 1-public) | |
| PCC dimension 5: Relationship between patient and health worker | **Desired characteristics of health workers** <br> **Considerate** *"The nurses who treat us (patients living with diabetes) are very responsible, they know that we are not allowed to eat before the morning tests and they always come on time and much earlier than the other health workers, we appreciate that"* (special clinic for diabetes patient, male, 49yrs, FGD facility1- public) <br> **Desired characteristics of patients** <br> *"You have to present yourself well when you go to the facility and carry all the things that you will need"* (MCH patient, female 31yrs, facility 3-public) <br> *"We also need to say thank you and appreciate our health workers, then you will always get good service, sometimes I also bring some food from the farm and they are very happy"* (MCH patient, male, 36yrs, facility 1-public) | **Undesired characteristics of health workers** <br> **Late and hurried** *"Here we always have to wait for a long time sometimes until 11. . .and when they come, they want to leave early, this can make you angry if you have been waiting and at the end of the day, you are told to come tomorrow"* (MCH patient, female 26years, facility 2-public) <br> **Undesired characteristics of patients** *"Sometimes we patients also go the facility when we are dirty and that can scare health workers, then they fail to take good care of us. Now when I have to go to the clinic, I make sure I am clean"* <br> *"To add to that some patients are also very rude, angry or showing off to the health workers, so the health workers avoid them, we can't blame the health workers all the time"* <br> (FGD facility 1-public) <br> *"in the labor ward, they don't like it if you cry a lot"* (MCH patient, female, 22yrs, facility 4-private) |

providing a private setting for consultation. Patients' did not feel appreciated when health workers were late and were less likely to express their thoughts about their illness if the health worker was harsh, or did not answer their questions. For instance, patients who felt that the

health worker was rude during their first interaction, were less likely to ask what the result of a laboratory test was. Another aspect that affected decision making was the patients understanding of what tests were done, the results and the treatment given. While a majority of them did not understand what the results of their lab tests were, they did not ask for an explanation out of fear of being reprimanded for not knowing or not understanding. However, patients did know how to take the medication they had been given. Most patients getting routine care services were familiar with paracetamol tablets, but had little understanding of other types of medication. Patients attending HIV/AIDS clinics in both private and public facilities understood the dosage and purpose of the anti-retroviral or TB drugs that they were taking, and also received advice on diet and sexual health. The adverse effects of medications were never discussed (see Box 3 section on finding common ground).

All the patients interviewed at the facility seeking curative care, when asked about health promotive messages, indicated that they got this type of information from VHTs (information on reproductive health), local council members (sanitation), the health workers (individual conditions) and sometimes radio announcements on public health campaigns and free medical camps. One of the patients also greatly appreciated getting visited at home when they could not make it to the health facility to pick their medication. One of the patients receiving specialized care for diabetes also expressed the importance of his patient group in encouraging him to keep healthy (see Box 3 section on health promotion).

The *relationship between the health worker and the patient* showcases that the health workers hold more power than the patient because of their position and their ability to provide the patient with a diagnosis and treatment. Culturally, women and children also knelt before the health worker as a sign of respect furthering the power differences. The relationship between a patient and health worker was influenced by characteristics of both patients and health workers. The relationship was perceived as better when the health worker was considerate, competent, spoke kindly and could be trusted to keep the patient's information confidential. Patients did not absolve themselves of all responsibility in communication citing that they too needed to be willing and open to communicate, present themselves as neat and clean at the facility, and be polite and treat the health workers with respect. Some patients also expressed the of appreciating the care they receive by thanking the health worker and sometimes even giving them rewards like food from the farm as an important part of maintaining good relationships with the health workers (see Box 3 section on the relationship between patients and health workers).

## Relating patient experiences to patient reported outcomes

In this section we describe whether patients' perception of their care experience influenced their perceived satisfaction and enablement. The first outcome of interest was patient-reported satisfaction. Univariate regression of patient satisfaction with PCC dimensions as independent variables resulted in a significant association for all PCC dimensions. However, when the regression was adjusted for all variables and interaction effects, only PCC dimensions 1 and 5 were significantly associated with patient satisfaction. On a scale of 1 to 100, a one-point increase in the score of PCC dimension 1—exploring health and illness, is coupled with an average 0.166 point increase in the patients' perception of satisfaction. Likewise, a 0.663 increase for every one-point increase in the score of PCC dimension 5—relationship between the patient and the health worker. On average, satisfaction is 3.397 points higher in private facilities, but the type of facility also interacts with the effect of PCC dimension 5 in that it is present in public facilities (0.663 points) but nonexistent in private facilities (-0.017 points; see Table 7 below). Patient-reported satisfaction was not significantly associated with the patients' gender, literacy levels or the type of service received. Univariate regression of patient-reported

**Table 7. Factors affecting patient reported satisfaction.**

| PATIENT SATISFACTION | | | |
|---|---|---|---|
| Descriptions | Univariate regression | Multivariate Regression (Adjusted to PCC dimensions) | Multivariate Regression (adjusted for all variables) |
| | Coefficient (95% CI) | Coefficient (95% CI) | Coefficient (95% CI) |
| | P value * P<0.05, | P value * P<0.05, | P value * P<0.05, |
| | ** P<0.01, | ** P<0.01, | ** P<0.01, |
| | *** P<0.001 | *** P<0.001 | *** P<0.001 |
| **Patient experience according to patient-centered dimensions** | | | |
| PCC1: Exploring health and the illness experience | 0.181 (0.122–0.240)*** | 0.112 (0.048–0.176)** | 0.166 (0.076–0.260)** |
| PCC2: Understanding the whole person | 0.061 (0.007–0.115)* | -0.034 (-0.096–0.028) | -0.122 (-0.323–0.079) |
| PCC3: Finding common ground | 0.103 (0.056–0.151)*** | 0.041 (-0.016–0.097) | -0.043 (-0.147–0.060) |
| PCC4: Prevention and health promotion | 0.059 (0.034–0.083)*** | 0.020 (-0.006–0.046) | 0.321 (-0.002–0.643) |
| PCC5: The patient and health worker relationship | 0.333 (0.232–0.435)*** | 0.262 (0.152–0.373)*** | 0.663 (0.452–0.873)*** |
| **Type of facility** | | | |
| Facility is a Public PHC | Ref | – | Ref |
| Facility is a Private PHC | 3.397 (2.274–4.521)*** | – | 29.249 (15.206–43.291)*** |
| **PCC Interaction effects (facility is public or private)** | | | |
| PCC1 Interaction effect | – | – | -0.116 (-0.270–0.038) |
| PCC2 Interaction effect | – | – | 0.268 (-0.038–0.574) |
| PCC3 Interaction effect | – | – | 0.132 (-0.012–0.277) |
| PCC4 Interaction effect | – | – | -0.278 (-0.788–0.232) |
| PCC5 Interaction effect | – | – | -0.680 (-1.023 –-0.338)*** |
| **Other patient/interviewee factors affecting reported patient satisfaction** | | | |
| **Age** | | | |
| 16–24 years | Ref | | Ref |
| 25–44 years | 1.511 (0 .137–2.886)* | – | 0.443 (-0.824–1.710) |
| Above 45 years | 0.278 (-1.408–1.963) | – | 0.412 (-1.222–2.047) |
| **Gender** | | | |
| Gender is Male | Ref | | Ref |
| Gender is Female | 0.101 (-1.329–1.530) | – | 0.566 (-0.756–1.887) |
| **Type of service received** | | | |
| Routine care | Ref | | Ref |
| Maternal and child health care | -0.35 (-1.777–1.077) | – | -0.673 (-1.956–0.611) |
| Specialized clinic | 0.31 (-1.117–1.737) | – | -0.104 (-1.463–1.255) |
| **Patient literacy (Reading English)** | | | |
| Cannot read English | Ref | | Ref |
| Can read English | 0.858 (-0.344–2.060) | – | 0.372 (-0.784–1.528) |

PCC: Patient-centered care

Ref: reference category.

satisfaction with patients' age as the independent variable revealed slightly significant association for patients aged 25–44 years, however this association was not significant after controlling for the effect of other variables on patient-reported satisfaction.

The second outcome of interest was patient enablement. Univariate regression of patient enablement with PCC dimensions as independent variables revealed significant associations with four PCC dimensions, with the patient and health worker relationship as the exception. However, after adjusting for all PCC dimensions, only PCC dimensions 2 and 3 were significantly (but weakly) associated (0.064 and 0.046 points, respectively). Patient enablement did

**Table 8. Factors affecting patient reported enablement.**

| PATIENT ENABLEMENT | | | |
|---|---|---|---|
| Descriptions | Univariate regression | Multivariate Regression (Adjusted to PCC dimensions) | Multivariate Regression (adjusted for all variables) |
| | Coefficient (95% CI) | Coefficient (95% CI) | Coefficient (95% CI) |
| | P value * P<0.05, | P value * P<0.05, | P value * P<0.05, |
| | ** P<0.01, | ** P<0.01, | ** P<0.01, |
| | *** P<0.001 | *** P<0.001 | *** P<0.001 |
| **Patient experience according to patient-centered dimensions** | | | |
| PCC1: Exploring health and the illness experience | 0.033 (0.001–0.067)* | 0.006 (-0.028–0.041) | 0.007 (-0.039–0.053) |
| PCC2: Understanding the whole person | 0.084 (0.057–0.112)*** | 0.064 (0.030–0.098)*** | 0.074 (0.028–0.120)** |
| PCC3: Finding common ground | 0.077 (0.052–0.101)*** | 0.046 (0.015–0.077)** | 0.045 (0.000–0.089)* |
| PCC4: Prevention and health promotion | 0.021 (0.008–0.034)** | 0.010 (-0.005–0.024) | -0.008 (-0.026–0.011) |
| PCC5: The patient and health worker relationship | 0.036 (-0.021–0.094) | -0.059 (-0.120–0.001) | -0.027 (-0.105–0.052) |
| **Type of facility** | | | |
| Facility is a Public PHC | Ref | – | Ref |
| Facility is a Private PHC | -0.089 (-0.721–0.543) | – | 2.377 (-5.683–10.437) |
| **PCC Interaction effects (facility is public or private)** | | | |
| PCC1 Interaction effect | – | – | 0.025 (-0.063–0.114) |
| PCC2 Interaction effect | – | – | -0.054 (-0.229–0.121) |
| PCC3 Interaction effect | – | – | -0.008 (-0.091–0.075) |
| PCC4 Interaction effect | – | – | 0.349 (0.056–0.642)* |
| PCC5 Interaction effect | – | – | -0.123 (-0.320–0.074) |
| **Other patient/interviewee factors affecting patient reported enablement** | | | |
| **Age** | | | |
| 16–24 years | Ref | | Ref |
| 25–44 years | 0.876 (0.148–1.604)* | – | 0.653 (-0.074–1.380) |
| Above 45 years | 1.166 (0.273–2.059)* | – | 0.463 (-0.476–1.401) |
| **Gender** | | | |
| Gender is Male | Ref | | Ref |
| Gender is Female | -0.927 (-1.680 –-0.174)* | – | -0.433 (-1.192–0.325) |
| **Type of service received** | | | |
| Routine care | Ref | | Ref |
| Maternal and child health care | -0.55 (-1.29–0.191) | – | -0.266 (-1.003–0.471) |
| Specialized clinic | 0.89 (0.149–1.631)* | – | 0.035 (-0.745–0.815) |
| **Patient literacy (Reading English)** | | | |
| Cannot read English | Ref | | Ref |
| Can read English | 0.523 (-0.115–1.161) | – | 0.451 (-0.212–1.115) |

PCC: Patient-centered care

Ref: reference category.

## Box 4. Patient suggestions for improving the quality of care that they receive

| Suggestion | Exemplifying Quote |
|---|---|
| Improve availability of medicine | *I wish they would improve in the stock of medicine, sometimes you come here, wait all day and you have to back home still sick and without medication* (routine care patient, Female, 30 years, facility 2-public) |
| Provide avenues for patients to file complaints and act on them | *"I want government to continue helping us and improve on the management of the facility, and to address our complaints."* (specialized clinic for diabetes patient, Male, 60 years, facility 1-public) |
| Improve availability of staff | *"To add the staff quarters so that we can have someone staying here at night and on weekends, it is very far to go to the district hospital if the basawo (health workers) are not here"* (routine care patient, Female, 43 years, facility 3-public) |
| Facility infrastructure and transport for referral | *"We need enough seats in the waiting space. . . and we also need transport to the district hospital. . .Cleaner toilets"* (MCH patient, Female, 29years, facility 4-private) |
| Home visits / home based care | *"I would also like if sometimes the health workers can be able to come to our homes, especially when you are so sick that you can't even get up to go to hospital, I wish the health worker would come home and see me"* (specialized clinic for ART patient, Female, 53years, facility 4-private) |

not differ significantly between public and private facilities, though there is a significant interaction effect whereby dimension 4 is not associated with enablement in public facilities (-0.008) but is associated with enablement in private facilities (0.341; see Table 8 below). Patient-reported enablement was not significantly associated with the patients' literacy levels. Univariate regression of patient enablement with patients' age, gender and the type of service received, as the independent variable revealed slightly significant association for patients aged 25–44 years, patients aged above 45years, female patients and patients receiving care in specialized clinics. However these associations were not significant when controlling for the effect of other variables on patient-reported enablement.

These separate analyses show how different patient reported outcomes are affected by PCC dimensions, their interactions, and other contextual factors that could affect patient perceptions of care and patient-centeredness. Lastly, patients also gave suggestions of what they would want to see improved in the facilities they visit in order to receive better services (see Box 4).

## Discussion

Incorporating patients' views in evaluating the care they receive has continued to gain momentum in the last decade with the goals of (1) involving patients in creating a demand for health information, (2) establishing patient preferences and values, (3) providing avenues where patients can give feedback on the services that they receive, and (4) participation in decision making at individual and health policy levels [48]. This discussion is organized according to the proposed conceptual framework (see Fig 1). We begin by considering factors that shape patient expectations before seeking health care. Secondly, we explore how patient expectations and experience of care (interpersonal aspects related to PCC) affect patient reported outcomes

of satisfaction and enablement. Additionally, we talk about our experience in measuring patient perceptions of care and patient-centeredness.

Before getting into a more detailed discussion on the quality of care, and more in particular its patient-centered character, offered in public and private facilities, it is important to note that our results revealed a number of important differences in the characteristics of patients seeking care at public or at private facilities. Patients using private facilities are relatively younger and more literate; the latter may indicate a higher socio-economic status. Quite noticeable also is that patients base their choice to use a public or a private facility on different considerations. Patients using public facilities say to do so because of their proximity and the absence of fees charged at the time and point of use. Patients using private facilities, on the other hand, say to do so because of their perception of better quality of care and higher responsiveness to patient needs and preferences like the availability of resources (health personnel, medication,, etc.), flexible opening hours, easier access to health workers, shorter waiting times, better organization of transport in case of referrals etc. We will discuss further whether these differences also translate in differences in outcomes of care.

## Factors shaping patients' health needs and expectations

One of the important aspects of PCC is offering care that is tailored to patients' values, health needs, and preferences [40, 41]. Literature on the conceptualisation of PCC by different actors in the US health system also points out that patients are viewed as people with a "compromised physiological state" and a "threatened identity" which makes them vulnerable [49]. PCC aims to give patients a sense of identity and control by making patients responsible for their own health and empowering them to be able to choose to participate in making decisions about their health [50]. Our study findings show that patients in Uganda perceive healthcare as something to be sought only when ill. This brings to the forefront the need for community engagement in preventive healthcare to include health promotion messages in addition to already existing vaccination and reproductive campaigns. Patients also preferred to seek healthcare at facilities that were accessible in terms of cost, distance and reliable in the delivery of services (including availability of staff and medication) even when facility is closed. This could explain the greater satisfaction in patients attending private facilities where their expectations of getting treatment were better met. Although care is free in public facilities, patients were worried about indirect costs of purchasing drugs or getting tested outside the facility–an ongoing concern in the provision of primary health care (especially for chronic illness) in LMICs [51]. These findings are also congruent with research done on patient views about the quality of primary care in Jordan—an upper middle income country—where constructs related to costs and delivery of service had the most positive impact on overall satisfaction of patients [52]. In Uganda, patients in both public and private facilities expressed that they prefer health workers who are compassionate and listen. However, more emphasis was placed on the health workers ability to keep patient information confidential, and patients in both public and private facilities in Uganda were not very keen on the interpersonal aspects of the interaction between them and their health workers (taking account of patients' preferences, considering psychological and social aspects of health and illness, and involving them in decisions). This is different from research done in high-income countries about gaining patient views about their consultations where patients value the interpersonal aspects of care just as much as the technical competence of the health worker [21, 42, 48, 53, 54]. This means that part of taking into account patient preferences involves letting them choose how much they want to be involved. A literature review on patients perception of quality in sub-Saharan Africa also

cautions that patients' views vary with subject recruitment site, depending on a rural or urban location and the socio-economic characteristics of the population, and recommends improvement in the methods used to examine patient views on quality of primary health care [55]. While most of the aforementioned studies recommend the training of health workers in communication and strengthening their skill in initiating interpersonal aspects of care, it is important to acknowledge that patients also need to be empowered to expect, even demand information and engaged to "participate in care in a way uniquely appropriate to the individual, in cooperation with a healthcare provider or institution, for the purposes of maximizing outcomes or improving experiences of care" [56].

## Relating patient perceptions of care and patient-centeredness to patient reported outcomes

Regarding dimensions of PCC, most patients reported higher scores with exploring health and the illness experience, and the patient and health worker relationship; but there were significantly lower scores regarding understanding the whole person, finding common ground / shared decision making, and health promotion. Awareness about the impact of psycho-social challenges to health care for both patients and health workers is crucial to understanding the patient as a whole person, a key component of PCC [40]. Our findings clearly show that a larger number of patients did not expect to receive psycho-social care from their health workers and did not discuss domestic (social), cultural, spiritual or psychological issues during consultations. They relegated these conversations to elders, families and friends. Previous studies in the same context also reveal the gaps in the Ugandan health system when it comes to providing PCC to patients with chronic illnesses who face a substantial psychosocial burden; and lack of insight in how patient-provider interactions affect processes of care especially for patients with psychological distress [57, 58]. Uniquely, patients receiving specialized care for chronic illnesses had more positive responses regarding understanding the whole person. This could be because patients with chronic illnesses have become more knowledgeable of the management of their health and illness promoted by long-term interaction with their health workers [59]. In addition, specific ('vertical') disease-control programmes related to specialized care for diseases like HIV/AIDS have received more attention over the years with programmes pumping in more resources, and more pressure to perform that is driven by incentives [60]. This may result in small islands of high performance and quality of care. Considering that most patients in our study perceived their relationship with the health worker as positive, quality improvement efforts could focus more on educating both patient and health workers on psycho-social aspects of health and healthcare, and providing a safe environment where these issues can be discussed.

In PCC, the patient is seen as an equal partner, able to participate in decision-making about his/her health care as evidenced by conceptual frameworks developed in Europe and Canada [54, 61]. This means that the patient is knowledgeable about his/her health [62–64] and cognizant of his/her right to privacy, autonomy and demand to be treated with dignity [63, 65]. In our study, this was not the case with most patients reporting difficulty in asking questions and low awareness of their rights. Most patients were comfortable with exchanging information about their symptoms with the health worker, and would have liked to be told about treatment options, but a smaller number of patients did not wish to be involved in discussing treatment options and preferred to leave the final decision to the health workers. This could be due to the perception that the health worker is most knowledgeable about health issues. Nevertheless, if health workers offered more information and opportunities to ask questions, patients reported feeling more comfortable to ask what is written in their patient book or what they don't

understand. This is corroborated by studies in contexts where PCC is more developed, they highlight the importance of both patient and health worker autonomy, and the patient-provider relationship as a fundamental element of shared decision making [66] which, in turn, then leads to positive primary health care outcomes [13]. Literature on research agendas involving patients in the Netherlands also emphasises the necessity of health organizations that are supportive towards involving patients in health policy making and evaluations [67]. Patient-centered interventions in contexts like Uganda need to take into account that capacities for shared decision making of patients, health workers and organizations need to be developed as part of, if not before the implementation of PCC.

Patients perception of what can be done to improve the quality of care was also driven by their preferences related to accessibility (financial, geographical and availability of health workers, and smooth referrals to receive care in higher level district hospitals) and reliability of services. None of the patients seemed aware of their own role in improving the quality of care that they receive. With all this in mind, patients in Uganda need to be encouraged to take up responsibility for their health, empowered with knowledge, and supported (including protection against the consequences of asking too many questions) in order for them to be able to participate in decision making and evaluating the care that they receive. This is supported by research on improving patient experiences with nurses in Australia, which prompts for health workers to provide an environment that encourages and build the patients' capacity to have control over their health and that result in positive patient experiences [68]. Both patients and health workers need to be aware of, appreciate and support each other's roles in patient education, empowerment and the provision of PCC.

## PCC and building resilient community health systems

As mentioned in the previous section, our findings show that most patients are yet to be empowered enough to participate in making decisions about health care due to the perceived information asymmetry and underlying power differences between health workers and patients. One way to overcome these challenges would be to strengthen the consumer voice by not only focusing on the patient as an individual agent of change, but by putting more emphasis on the capabilities of the community as a whole. This can be achieved by combining the voices of patients, patient support groups, VHT members / community health workers and local administrators to form more comprehensive and influential community health systems [69]. Pfaffmann *et al.* further emphasize that "*Universal health coverage in all countries by 2030 is unattainable without strengthening community health systems —enabling community health workers to deliver preventive and curative services, and supporting the empowerment of communities to demand social accountability from their governments and other providers for coverage of quality health services*" [70]. In Uganda, community meetings with the intention to improve the relationships, and promote mutual trust between patients and VHTs could be an additional step. Furthermore, research on the performance of community health workers convey that common understanding, trusting relationships and balanced power between different actors in the health system are essential to the functioning of community health systems within the wider complex adaptive health system [71]. Strengthening the links with civil society and the Uganda National Health Consumers Organization in particular, would empower patients and communities to effect positive change—in both the implementation of PCC and the quality of care they receive—by providing awareness of patient rights and offering avenues where patients file their complaints, and make recommendations that are considered by health providers without unintended consequences to the patient.

## Measuring patient perceptions of patient-centered care

Many models are recommended for measuring patients' perceptions of individual aspects of PCC [43]. Using these models we developed a conceptual framework, which enabled us to look at the patients' expectations *before* the consultation, *during* the experience of care and perceived outcome *after* the consultation. This framework is useful in measuring patients' perceptions of PCC during a particular visit to the facility, but would be challenging to apply in measuring the change of patient perceptions over time. Due to the focus on interpersonal aspects of health, it does not consider clinical procedures during consultations and further modifications would be required to accommodate clinical outcomes and adherence. Using our framework, we were still able to study the impact of patient experience on outcomes. The study findings show that satisfaction is more associated with the patients experience in exploring health and the illness experience, and with the patient and health worker relationship. Patient enablement on the other hand is more associated with the patients experience in understanding the whole person and finding common ground / shared decision making. After adjustments to incorporate other variables, differences in satisfaction were very wide between public and private facilities, while patient enablement differences remained stable. This could indicate that patient enablement would be a better measure for improvements in PCC related outcomes in similar contexts, and provide a guide to more targeted resource allocation. Notably, attributes of patient enablement are very similar to attributes of PCC including the "consideration of the whole person (with physical, emotional and psychosocial needs), therapeutic relationship, the facilitation of learning, the valorization of the person's strengths, the implication and support to decision making and the broadening of the possibilities" [72]. Similar studies on routine consultations in primary care also recommend the use of patient enablement as an outcome measure [42, 73].

## Recommendations for further research

While our research looked at patient perceptions related to PCC in the primary care context, further research is recommended to better understand the relationship between PCC aspects to both patient-reported and clinical outcomes long-term. Research is also required to explore the implementation of PCC at higher levels of care, including emergency care. We also need to acknowledge that patient perceptions of quality care vary across countries [74], can change over time, and depend on the type of care that patients receive [75]. Patient perceptions are therefore affected by other patient specific and contextual factors (past experiences, frequency of visiting the facility, distance from the facility, waiting time, resources available at the facility etc.). In addition, there are many areas of overlap in measuring the similar concepts of PCC, patient enablement, patient empowerment, decision-making and community participation respectively; particularly in resource constrained contexts [76]. Models that take these complexities into consideration may provide more accurate means to compare PCC in different contexts and patient populations.

## Implications for policy and practice

Our findings show that patients in Uganda have some understanding of specific concepts related to PCC, and express a demand for it. This provides additional evidence to support the inclusion of PCC in the quality improvement agenda. Our description of how specific PCC dimensions relate to patient reported outcomes could offer a starting point for small scale patient-centered interventions in resource constrained settings. However, in order for PCC interventions to progress, patients and health workers need to be involved from the design stages, stakeholders should also be involved in recommending contextualized indicators of

progress and participate in the evaluation of interventions. However, before all the above can be accomplished, patients need to be empowered and have a shared vision with other stake-holders on PCC [39]. Training on interpersonal aspects of health care and provision of incentives to health workers to encourage psycho-social care provision could be a worthwhile avenue in promoting the implementation of PCC. Last but not least, developing and validating methods to measure PCC in similar contexts would be of great value in monitoring progress and impact of PCC on outcomes.

## Strengths and limitations of this study

This study presents an analysis of patient perceptions and experience with different dimensions of PCC and relates it to the outcomes of patient satisfaction and enablement. As with most satisfaction surveys, social desirability response bias is a concern [77]. We have attempted to counter this effect by providing two measurements of outcome and confirming the findings using qualitative results. We acknowledge that the majority of our interviewees were female–this could be due to the intentional inclusion in our study a of the specific category of patients/caregivers seeking and offering maternal and child health care. Both groups are predominantly female. Future studies should consider studying gendered perspectives of PCC.

Our study design was cross-sectional and further research is thus required to identify differences in patient perceptions over time. Finally, another limitation was the normalization of Likert scale data for the descriptive statistical analysis, which assumes that the degrees of difference between the statements 'strongly agree', 'agree', 'strongly disagree' and 'disagree' would be equal. Acknowledging the ongoing debate in using parametric tests for ordinal variables [78], we aimed to make the results more robust by using Likert items that had been validated in developed countries, engaging into appropriate translation and back translation of each item, contextualizing words and phrases with different meanings, pilot testing the tools, using a sizable sample size, and analyzing actual Likert scale scores (in addition to individual items). We also triangulated quantitative findings with qualitative findings to better answer our research questions [79].

## Conclusion

PCC is gaining increasing attention as an approach to improve the quality of care provided in LMICs. Empirical evidence suggests that there is a demand for patients to be more involved in maintaining their health, and improve health systems accountability by participating in evaluating the healthcare that they receive. In practice, our findings show the need to be cognizant of the unique challenges in resource constrained settings. To start with, patients' expectations and perceptions of good quality health care are still largely driven by biomedical and technical aspects, more than psychosocial or interpersonal aspects of care. Patients may not be eager to be involved because they do not know that the way how they feel about their health and health care is of great value, and that it is their right to participate. Targeted health education on patients' responsibility for their health, and creating awareness of their rights are essential in building patients' confidence to participate in PCC.

Regarding the experience of PCC, our findings show that less attention was given by both patients and health workers to understanding the whole person (psychosocial aspects of health) and shared decision making. Existing power dynamics and information asymmetry further inhibit shared decision making between health providers and patients. Implementation of PCC would require capacity building not only for patients and communities to be more confident to share their perspectives, but also involve health workers to promote dialogue and create decision-making spaces where patients are viewed and involved as equal partners. Our

comparison of the differences in the perception of PCC between patients attending public and private facilities also indicate that there are important differences in the characteristics of patients seeking health care in the respective facilities and in how they view quality, which in turn can affect their perception of the care experience and influence patient reported outcomes.

Lastly, our use of two key patient reported outcomes (patient satisfaction and patient enablement) shows how patient perceptions of PCC dimensions can affect patient reported outcomes disparately. Further research is therefore recommended to develop and validate methods used in the implementation and measurement of PCC, especially in resource-constrained contexts. This is important in order to track the impact of PCC on not only patient reported outcomes, but also on populational health outcomes over time.

## Supporting information

**S1 Appendix. Translated tools to measure patient perceptions of PCC in Uganda.** A compilation of the informed consent forms and tools used in qualitative and quantitative data collection (both in English and translated into Lusoga).
(DOCX)

**S1 Table. Patient-centered care measurement instruments used in designing the tool to measure patient perceptions of PCC in Uganda.** A table showing how the questions incorporated in the tools used in the study were developed from previously validated tools and supporting literature.
(DOCX)

**S2 Table. Patients' perceptions of patient-centered care dimensions and patient reported outcome measures, in rural eastern Uganda.** A table showing the responses of 300 patients to questions about their experience of PCC, and patient reported outcomes.
(DOCX)

**S1 Dataset. Patient exit interview dataset.** A STATA dataset with anonymized and de-identified patient data and responses.
(DTA)

## Acknowledgments

We would like to acknowledge the Makerere University School of Public Health and the Iganga-Mayuge Health and Demographic Surveillance Site for their assistance in the recruitment of study participants; our local field assistants and data collectors Judith Bikobere, Paul Waiswa, Betty Kyobe, and Gladys Sanga; the health workers who hosted us in their health facilities; village health team members and all the women and men who participated in this research.

## Author Contributions

**Conceptualization:** Everlyn Waweru, Joanna Orne-Gliemann, Freddie Ssengooba, Jacqueline Broerse, Bart Criel.

**Data curation:** Everlyn Waweru, Freddie Ssengooba, Jacqueline Broerse, Bart Criel.

**Formal analysis:** Everlyn Waweru, Tom Smekens, Bart Criel.

**Funding acquisition:** Everlyn Waweru, Jacqueline Broerse, Bart Criel.

**Investigation:** Everlyn Waweru, Jacqueline Broerse, Bart Criel.

**Methodology:** Everlyn Waweru, Joanna Orne-Gliemann, Freddie Ssengooba, Jacqueline Broerse, Bart Criel.

**Project administration:** Everlyn Waweru, Freddie Ssengooba, Jacqueline Broerse, Bart Criel.

**Resources:** Everlyn Waweru, Jacqueline Broerse, Bart Criel.

**Software:** Tom Smekens.

**Supervision:** Everlyn Waweru, Joanna Orne-Gliemann, Freddie Ssengooba, Jacqueline Broerse, Bart Criel.

**Validation:** Everlyn Waweru, Tom Smekens, Bart Criel.

**Visualization:** Everlyn Waweru, Tom Smekens.

**Writing – original draft:** Everlyn Waweru, Bart Criel.

**Writing – review & editing:** Everlyn Waweru, Tom Smekens, Joanna Orne-Gliemann, Freddie Ssengooba, Jacqueline Broerse, Bart Criel.

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
