## [Decision Letter · Decision Letter 0]

1 Jun 2020

PONE-D-20-10068

Patient perspectives on interpersonal aspects of healthcare and patient-centeredness at primary health facilities: a mixed methods study in rural Eastern Uganda

PLOS ONE

Dear Dr. Waweru,

Thank you for submitting your manuscript to PLOS ONE. After careful consideration, we feel that it has merit but does not fully meet PLOS ONE’s publication criteria as it currently stands. Therefore, we invite you to submit a revised version of the manuscript that addresses the points raised during the review process.

We look forward to receiving your revised manuscript.

Kind regards,

Elena Ambrosino

Academic Editor

PLOS ONE

Journal Requirements:

Reviewers' comments:

Reviewer's Responses to Questions

**Comments to the Author**

1. Is the manuscript technically sound, and do the data support the conclusions?

Reviewer #1: Yes

Reviewer #2: Yes

2. Has the statistical analysis been performed appropriately and rigorously? 

Reviewer #1: Yes

Reviewer #2: I Don't Know

3. Have the authors made all data underlying the findings in their manuscript fully available?

Reviewer #1: Yes

Reviewer #2: Yes

4. Is the manuscript presented in an intelligible fashion and written in standard English?

Reviewer #1: Yes

Reviewer #2: Yes

5. Review Comments to the Author

Reviewer #1: Well written paper on an important aspect of health care which I think should be accepted.

I just have a couple of comments:

The result sub-sections (p17,21 and 28) would be improved by using the same titles as used in the introductory list on page 16.

Generally small numbers (ie less than 5) are not included in summary tables because of confidentiality and reliability. Suggest either collating the responses or replacing the small numbers with <5.

Need to consistently abbreviate patient centered care. Sometimes abbreviated and sometime provided in full.

The discussion although generally relevant to the paper could be written more concisely to lead to the conclusions.

Reviewer #2: This is a mixed methods paper considering patient perspectives on interpersonal aspects of healthcare and patient-centerdness in Uganda.

This is a well presented manuscript.

In the abstract I was less clear about the first sentence (line 30/31) – the results section in Abstract could be strengthened for impact and clarity.

Some statements are made in the manuscript that need to be backed up by the literature e.g. line 57-58. It would also be helpful to distinguish between the Countries of origin of studies. You nicely situate the Ugandan interventions.

Characteristics of the healthcare facilities (Table 1) seems a little out of place in methods section. Can perhaps move to results or provide as a supplementary table from Introduction.

Really nice Figure 1 and Figure 2.

You do not highlight male respondent numbers as a limitation. For this paper it may be better to focus only on the female respondents and orient the paper findings wholly to female aspects. If kept in, please discuss how respondent numbers of males might be so low/why.

Reference 2 and 21 – please check spelling of WHO name (Organization)

Data underlying the findings is not provided, some restrictions apply. A link is provided to the source of the data.

6. PLOS authors have the option to publish the peer review history of their article (what does this mean?). If published, this will include your full peer review and any attached files.

Reviewer #1: Yes: A/Prof Margo Barr

Reviewer #2: Yes: Lesley Gray

---

## [Author Response · Author response to Decision Letter 0]

22 Jun 2020

PONE-D-20-10068

Patient perspectives on interpersonal aspects of healthcare and patient-centeredness at primary health facilities: a mixed methods study in rural Eastern Uganda

To Elena Ambrosino,

Academic editor and editorial,

PLOS ONE,

Dear Elena and the PlosONE editorial team,

Re: Response to academic editor comments

Thank you for taking your time and using your resources to review our research article on “Patient perspectives on interpersonal aspects of healthcare and patient-centeredness at primary health facilities: a mixed methods study in rural Eastern Uganda”. 

The co-authors and I have addressed the comments to the best of our ability. In the revised submission, please find attached:

• A letter responding to each point raised by the academic editor and reviewers, labelled 'Response to academic editor and reviewers'.

• A marked-up copy of the manuscript that highlights changes made to the original version labelled 'PONE-D-20-10068 Revised Manuscript with Track Changes'.

• An unmarked version of the revised manuscript without tracked changes, labelled 'PONE-D-20-10068 Manuscript'.

I would like to confirm that there are no changes to the financial disclosure section. The study was supported by funding from the European Commission, through the Erasmus Mundus Joint Doctorate Fellowship, Specific Grant Agreement 2016-1346, awarded to EW (Everlyn Waweru). We have now included this as a section in the manuscript after acknowledgements. I also have no laboratory protocols to deposit. However, as requested by the reviewers we now provide anonymised quantitative patient dataset as an additional supplementary file 4.

Many thanks and kind regards,

Everlyn Waweru

PhD Candidate - TGH Erasmus Mundus

Department of Public Health

Unit of Equity and Health

Mobile: +254722996857

ewaweru@itg.be ; evelynwaweru@gmail.com

 

Journal Requirements:

The format and file naming instructions have been updated

The anonymized and de-identified quantitative patient exit data set is provided as a supplementary file and quotes used are provided in the text. Interview audio files and transcripts are governed under the Institute of Tropical Medicine (ITM) data sharing and open access policy. Access requests for ITM research data can be made to ITM’s central point for research data access by means of submitting a completed Data Access Request Form. These requests will be reviewed for approval by ITMs Data Access Committee, with further approval from the ITM Research Ethics Committee. Please see this link for more information https://www.itg.be/F/data-sharing-open-access. Contact information for ITM data access committee is eb.gti@sseccaatadhcraeserMTI.

Review Comments to the Author

Reviewer #1: Well written paper on an important aspect of health care which I think should be accepted.

Thank you for your gracious comments and for taking your time to read and review our manuscript

I just have a couple of comments:

The result sub-sections (p17,21 and 28) would be improved by using the same titles as used in the introductory list on page 16.

Thank you for this observation, the sub titles have now been edited to reflect the sub-topics in the introduction to the result section.

Generally small numbers (i.e. less than 5) are not included in summary tables because of confidentiality and reliability. Suggest either collating the responses or replacing the small numbers with <5.

We have made the decision to replace values less than 5 with <5 for content in the manuscript and supplementary file 3. However, we retain the values less than 5 for the patient exit interview data set (supplementary file 4) to enable replication of analysis.

Need to consistently abbreviate patient centered care. Sometimes abbreviated and sometime provided in full.

We have abbreviated PCC in the text but left it in full 

• for the first mention in the abstract, first mention in the introduction, 

• in the sub-titles, figure titles, table titles, supplementary file titles and, 

• in instances where we are referring to patient centeredness or patient centered character where using the abbreviation may change the meaning. 

The discussion although generally relevant to the paper could be written more concisely to lead to the conclusions.

We have reworked the discussion, also shortening it. We hope that this section is now a better read, and matches the conclusion. 

Reviewer #2: This is a mixed methods paper considering patient perspectives on interpersonal aspects of healthcare and patient-centredness in Uganda.

This is a well presented manuscript.

Thank you for your kind words and comprehensive review.

In the abstract I was less clear about the first sentence (line 30/31) – the results section in Abstract could be strengthened for impact and clarity.

We have now re-written the results section to emphasize on the key findings of the study in the abstract, we also emphasise the structure of the results section of the abstract in line with the conceptual framework and result section of the manuscript.

Some statements are made in the manuscript that need to be backed up by the literature e.g. line 57-58. 

We have re-worded the section to present how historically, quality improvement strategies and assessments of primary health care performance in LMICs have focused on health care providers, with little or no attention to consumer perspectives. We have also now provided the accompanying references.

It would also be helpful to distinguish between the Countries of origin of studies. You nicely situate the Ugandan interventions.

In the discussion, we have now made clear the countries of origin of studies in cases where it is one particular country, but for reviews, we have not mentioned all the countries considered in the reviews but we mention the regions e.g. sub-Saharan Africa. 

Characteristics of the healthcare facilities (Table 1) seems a little out of place in methods section. Can perhaps move to results or provide as a supplementary table from Introduction.

Thank you for this suggestion, we have moved the characteristics of facilities to the first section of the results, therefore table 1 is now the summary of data collected and Table 2 the characteristics of facilities, we hope this improves the transition from the methods to the results section.

Really nice Figure 1 and Figure 2. 

Thank you we appreciate you recognising our efforts

You do not highlight male respondent numbers as a limitation. For this paper it may be better to focus only on the female respondents and orient the paper findings wholly to female aspects. If kept in, please discuss how respondent numbers of males might be so low/why.

We believe it would be disadvantageous not to include male patients and care givers because it would go against our random selection of patients. We hypothesise that the respondent numbers of males is low because in primary health care facilities in LMICs most care seekers and care givers would be female. Furthermore, the higher number of female interviewees may also be because of our sampling of patients seeking maternal and child health who would - as expected - be predominantly female. We included this in the limitations section, with the appropriate explanation.

Reference 2 and 21 – please check spelling of WHO name (Organization)

This has now been corrected

Data underlying the findings is not provided, some restrictions apply. A link is provided to the source of the data.

The anonymised and de-identified quantitative patient exit data set is now provided as supplementary file 4

---

## [Decision Letter · Decision Letter 1]

9 Jul 2020

Patient perspectives on interpersonal aspects of healthcare and patient-centeredness at primary health facilities: a mixed methods study in rural Eastern Uganda

PONE-D-20-10068R1

Dear Dr. Waweru,

We’re pleased to inform you that your manuscript has been judged scientifically suitable for publication and will be formally accepted for publication once it meets all outstanding technical requirements.

Kind regards,

Elena Ambrosino

Academic Editor

PLOS ONE

Reviewers' comments:

Reviewer's Responses to Questions

**Comments to the Author**

1. If the authors have adequately addressed your comments raised in a previous round of review and you feel that this manuscript is now acceptable for publication, you may indicate that here to bypass the “Comments to the Author” section, enter your conflict of interest statement in the “Confidential to Editor” section, and submit your "Accept" recommendation.

Reviewer #1: All comments have been addressed

2. Is the manuscript technically sound, and do the data support the conclusions?

Reviewer #1: Yes

3. Has the statistical analysis been performed appropriately and rigorously? 

Reviewer #1: Yes

4. Have the authors made all data underlying the findings in their manuscript fully available?

Reviewer #1: Yes

5. Is the manuscript presented in an intelligible fashion and written in standard English?

Reviewer #1: Yes

6. Review Comments to the Author

Reviewer #1: Well written manuscript. Authors have addressed all of the previously raised issues. No further issues need to be addressed.

7. PLOS authors have the option to publish the peer review history of their article (what does this mean?). If published, this will include your full peer review and any attached files.

Reviewer #1: **Yes: **A/Prof Margo Barr

---

## [Editor Report · Acceptance letter]

13 Jul 2020

PONE-D-20-10068R1 

Patient perspectives on interpersonal aspects of healthcare and patient-centeredness at primary health facilities: a mixed methods study in rural Eastern Uganda 

Dear Dr. Waweru:

I'm pleased to inform you that your manuscript has been deemed suitable for publication in PLOS ONE. Congratulations! Your manuscript is now with our production department. 

Kind regards, 

on behalf of

Dr. Elena Ambrosino 

Academic Editor

PLOS ONE